

# Investigating the limiting aircraft design-dependent and environmental factors of persistent contrail formation

Liam Megill[1, 2] and Volker Grewe[1, 2]

[1]Deutsches Zentrum für Luft- und Raumfahrt, Institut für Physik der Atmosphäre, Oberpfaffenhofen, Germany
[2]Delft University of Technology, Faculty of Aerospace Engineering, Section Aircraft Noise and Climate Effects, Delft, The Netherlands

**Correspondence:** Liam Megill (liam.megill@dlr.de)

**Abstract.** Mounting evidence has highlighted the role of aviation non-$CO_2$ emissions in contributing to anthropogenic climate change. Of particular importance is the impact of aircraft contrails and induced cloudiness, which recent studies attribute to over one third of the total Effective Radiative Forcing from aircraft operations. However, the relative importance of the aircraft design-dependent and environmental factors that influence and limit the formation of persistent contrails is not yet well

understood. In this paper, we use ERA5 data from the 2010 decade to better understand the interplay between the factors on a climatological timescale. We identify ice supersaturation as the most limiting factor for all considered aircraft designs, underscoring the importance of accurately estimating ice supersaturated regions. We also develop climatological relationships that describe potential persistent contrail formation as a function of the pressure level and Schmidt-Appleman mixing line slope $G$, and find that the influence of aircraft design on persistent contrail formation reduces with increasing altitude. Globally-averaged

persistent contrail formation could increase by 13.8 % for next generation conventional aircraft, or by 71.4 % if all aircraft were to be replaced by hydrogen combustion or fuel cell equivalents. On the other hand, water vapour extraction technologies such as the Water Enhanced Turbofan concept have the potential to reduce persistent contrail formation by 53.6 % to 85.6 %. The introduction of novel aviation fuels and propulsion technologies, therefore, present both challenges and opportunities with respect to persistent contrail formation.



## 1 Introduction

Aviation contributes to anthropogenic climate change through $CO_2$ emissions and non-$CO_2$ emissions and effects. Of particular importance is the formation of condensation trails, or contrails, which can form behind aircraft. Although most contrails quickly dissipate and have no effect on the climate, in certain ambient conditions contrails can spread to form contrail-cirrus clouds and persist for many hours (Haywood et al., 2009; Schumann and Heymsfield, 2017). Current best estimates suggest that

contrails and the resulting increase in aircraft-induced cloudiness could be responsible for around a half of the anthropogenic Effective Radiative Forcing in 2018 stemming from aircraft operations between 1940 and 2018 (Lee et al., 2021), although these estimates are highly uncertain (Kärcher, 2018; Lee et al., 2021; Burkhardt and Kärcher, 2011). As aircraft using alternative aviation fuels such as Sustainable Aviation Fuels (SAFs) and hydrogen ($H_2$) are proposed and developed, there is a need to more closely examine their potential contrail climate impact.

Contrails are formed when the hot and moist aircraft engine exhaust plume expands and cools. If, during expansion, the exhaust plume becomes supersaturated with respect to liquid water, water vapour condenses around ice nuclei in the exhaust, primarily soot (Kärcher and Yu, 2009; Kärcher et al., 2015). If the ambient temperature is below the homogeneous freezing temperature, the water droplets freeze to form ice crystals, creating a visible contrail. These conditions together constitute the Schmidt-Appleman Criterion (SAC, Schumann, 1996; Schmidt, 1941; Appleman, 1953). A lack of ice nuclei in the exhaust

does not prevent a contrail from forming or persisting: In low soot or soot-free conditions, for example behind aircraft using SAF or $H_2$ respectively, volatile exhaust particles and ambient, natural or anthropogenic aerosols can also activate into water droplets (Yu et al., 2024; Kärcher et al., 2015; Bier and Burkhardt, 2019; Kärcher, 2018). However, due to the large variability in the properties and concentration of the ambient aerosols (e.g. Brock et al., 2021; Voigt et al., 2022), the resulting ice crystal number and radiative effect of the subsequent contrails are currently highly uncertain.

Contrails can persist for many hours if the ambient conditions are supersaturated with respect to ice (Haywood et al., 2009). These contrails slowly sink and spread out due to wind shear, forming contrail cirrus and often mixing with natural cirrus. Contrail cirrus reflect incoming shortwave radiation and trap outgoing longwave radiation (Kärcher, 2018). The radiative effect of contrail cirrus is thus diurnal in nature, with contrails warming during the night and both warming and cooling during the day. However, on average, the warming effect has been shown to dominate (Meerkötter et al., 1999; Grewe et al., 2017b; Lee

et al., 2021).

Recent studies have shown that a small number of flights have an over-proportional contrail warming effect (e.g. Grewe et al., 2014; Teoh et al., 2020, 2022a). Avoiding contrail formation, in particular the so-called "big hits", is a topic of ongoing research (e.g. Sausen et al., 2024; Gierens et al., 2008; Filippone, 2015; Rosenow and Fricke, 2019). Several open questions should be addressed before contrail avoidance is routinely used in daily operations (Grewe et al., 2017c). One significant

hindrance to contrail avoidance schemes is that although contrail formation can be reliably forecasted, persistence currently cannot (Gierens et al., 2020; Wilhelm et al., 2022; Hofer et al., 2024b).

With this research, we take a more holistic approach to the persistent contrail formation, focusing on the development of longer-term strategies rather than individual avoidance. We aim to better understand the interplay between aircraft-dependent



and -independent factors which limit persistent contrail formation on a climatological timescale. Persistent contrail formation

can be limited by three main factors: droplet formation, droplet freezing and persistence. Our objectives are 1) to identify which factor(s) is/are generally responsible for the boundaries of persistent contrail formation regions; 2) determine the altitude-, latitude- and seasonal dependence of these factors and of persistent contrail formation regions; and 3) explore possibilities for the targeted introduction of future aircraft designs to reduce persistent contrail formation. This study is thus a step towards the development of a new, computationally inexpensive method to analyse the contrail climate impact of novel aircraft fuels and

propulsion technologies. Next to conventional kerosene, we consider the following technologies: Sustainable Aviation Fuels (SAFs), hydrogen ($H_2$) fuel cells and combustion, hybrid-electric aircraft and the Water-Enhanced Turbofan (WET) concept (Schmitz et al., 2021). We describe the potential contrail impacts of each technology briefly below.

SAFs have the potential to reduce the climate impact of aviation in the shorter term. The most obvious benefit of using such fuels is the reduction of lifecycle $CO_2$ emissions, but SAF usage may also affect the formation and climate impact of contrails.

SAFs tend to have a higher H/C ratio and lower proportion of aromatics compared to conventional kerosene. This results in a lower soot and ice crystal number concentration (Moore et al., 2017; Bräuer et al., 2021b; Voigt et al., 2021), which in turn reduces the optical depth and climate impact of contrails (Burkhardt et al., 2018). However, the higher H/C ratio also results in increased water vapour emissions and thus to the more frequent formation of persistent contrails (Rojo et al., 2015; Caiazzo et al., 2017; Hofer et al., 2024a; Teoh et al., 2022b).

In the longer term, another promising technology is hydrogen, either combusted in a jet engine or used in a fuel cell. The use of hydrogen has no direct $CO_2$ emissions and, unlike conventional jet fuels, is not expected to produce any soot or particulate emissions, although the formation of ultrafine volatile particles from lubricant oil vapours has been observed in lab combustion tests (Ungeheuer et al., 2022). The use of $H_2$ in a jet engine will also significantly increase water vapour emissions, from 1.25 kg/kg(kerosene) for conventional kerosene to around 8.94 kg/kg(hydrogen). Taking into account that $H_2$ has a gravimetric

energy density around 3 times higher than kerosene - 120.9 compared to 43.6 MJ/kg - the slope of the SAC mixing line is around 2.6 times larger for $H_2$ combustion than for kerosene for the same overall propulsion system efficiency. Given the low exhaust temperature, that factor is even higher for $H_2$ fuel cells, between 2.7 and 8.2 times, depending on the operating voltage of the fuel cell (Gierens, 2021). We can, therefore, expect a significant increase in contrail formation from $H_2$ aircraft (e.g. Grewe et al., 2017a; Kaufmann et al., 2024). However, recent studies (Sanogo et al., 2024; Wolf et al., 2023a) show that

contrail formation does not scale linearly with this factor. Kaufmann et al. (2024) in particular show that contrail formation from hydrogen-powered aircraft is strongly dependent on season, altitude and latitude. This suggests that the ambient atmospheric conditions and processes, rather than the aircraft design, mostly limit persistent contrail formation.

Other technologies reduce the slope of the mixing line, promising less frequent contrail formation - here we consider hybrid-electric aircraft and the Water-Enhanced Turbofan (WET) concept. Hybrid-electric aircraft are promising in that they could be

flown such that only electric power is used within regions of potential persistent contrail formation. Even when electric power is combined with conventional fuel during continuous use, the mixing line slope of the combined system is lower, resulting in fewer contrails forming (Yin et al., 2020). However, hybrid electric aircraft have a limited use case due to their low speed and service ceiling. Another possibility for reducing the mixing line slope is by condensing, storing and releasing water in areas in





which the ambient conditions are not conducive to persistent contrail formation. A notable example of this is the WET concept
(Schmitz et al., 2021; Pouzolz et al., 2021; Kaiser et al., 2022). The benefit of such a system is that it can be integrated into a
wider range of aircraft and is not limited by speed and service ceiling.

In the following section, we describe the methodology and data used in this study. The resulting limiting factors of climatological persistent contrail formation are analysed, discussed and compared to one another in Section 3. We conclude the study
with Section 4.

## 2   Data and methods

This section outlines the data and methods used to calculate the limiting factors of persistent contrail formation on a climatological basis. Section 2.1 describes the thermodynamic contrail formation and persistence criteria, and provides the calculation
methods for the mixing line slopes of various propulsion systems. We cover the full range of possible mixing line slopes, but
also specify certain aircraft designs for easier comparison between different technologies, which are described in Section 2.2.
We provide an overview of our air traffic scenario in Section 2.3; the European Centre for Medium-Range Weather Forecasts
(ECMWF) ERA5 dataset (Hersbach et al., 2020) and the corresponding humidity corrections used in this study in Section 2.4.
Finally, the limiting factors and maximum slope analyses are described in Sections 2.5 and 2.7 respectively.

### 2.1   Contrail formation and persistence

As described in the introduction, contrails form as a result of isobaric mixing of the hot and moist aircraft engine exhaust plume
with the cold and dry ambient air. The mixing process can be thermodynamically approximated by connecting the exhaust
and ambient conditions with a straight line on a temperature vs. water vapour partial pressure diagram, such as Figure 1 (gray
dashed lines). The slope $G$ of the mixing line is a function of the ambient pressure as well as aircraft engine and fuel-dependent
properties. For conventional kerosene (CON), the slope can be calculated using the following equation (Schumann, 1996),

$$G_{CON} = \frac{c_p p_a}{\epsilon} \frac{EI_{H_2O}}{(1-\eta)Q} \tag{1}$$

where $c_p = 1004$ J/kg/K is the isobaric heat capacity of air, $p_a$ the ambient pressure [Pa], $\epsilon = 0.622$ the molar mass ratio of
water vapour and dry air, $EI_{H_2O}$ the emission index of water vapour [kg/kg(fuel)], $\eta$ the overall propulsion system efficiency,
and $Q$ the lower heating value of the fuel [MJ/kg].

This definition was adapted by Yin et al. (2020) to parallel hybrid battery-electric aircraft by introducing a ratio $R$, where
$R = 1$ is pure liquid fuel operation and $R = 0$ is pure electric operation. $Q_E^0$ is defined as $Q(\eta_K/\eta_E)$, where $K$ refers to liquid
fuel (kerosene), and $E$ to the electric system.

$$G_{HYB} = \frac{c_p p_a}{\epsilon} \frac{R \cdot EI_{H_2O}}{R(1-\eta_K)Q + (1-R)(1-\eta_E)Q_E^0} \tag{2}$$





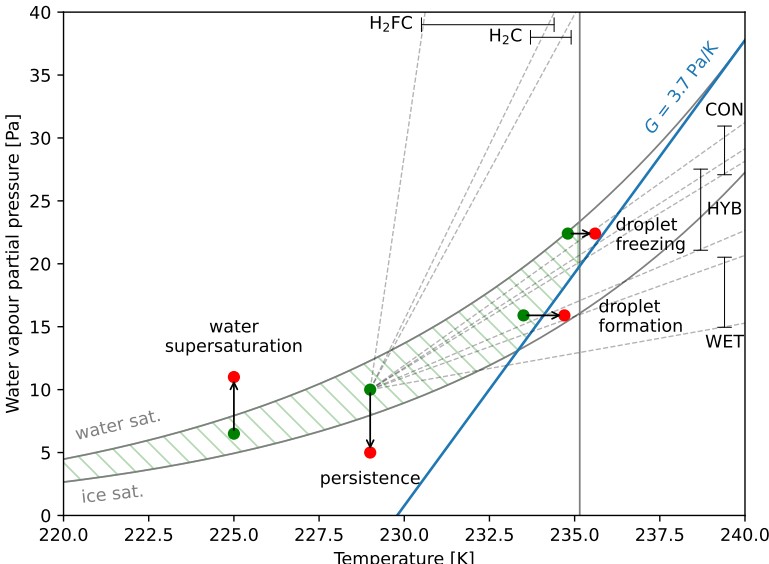

**Figure 1.** Ambient temperature vs. water vapour partial pressure diagram with an example threshold mixing line for an aircraft with $G = 3.7$ Pa/K. A persistent contrail would form behind this aircraft if the ambient conditions are within the green hashed area. Four limiting factors can be identified that define this area, represented by the arrows from green (persistent contrail forms) to red (persistent contrail does not form) points. Ranges of mixing line slopes for each aircraft type at 250 hPa are shown with dotted lines.

The definition was further modified for hydrogen combustion ($H_2C$) and fuel cell ($H_2FC$) aircraft by Gierens (2021). Eq. (3) is very similar to the standard definition and is used for hydrogen combustion, where $|\Delta h| = 120.9$ MJ/kg.

$$G_{H_2C} = \frac{c_p p_a}{\epsilon} \frac{EI_{H_2O}}{(1-\eta)|\Delta h|} \tag{3}$$

The calculation for fuel cells is more complex. Fuel cells operate in conjunction with electric motors, which fully decouples the exhaust from the propulsion. The exhaust could be modified to recover heat, separate and store water or achieve a desired ice crystal size. The properties of contrails produced by fuel cell powered aircraft thus depend on the design of the fuel cell system to a much larger degree than for traditional gas turbine engines. Moreover, due to the potentially very high supersaturations reached in the plume, homogeneous droplet nucleation could also take place even without the presence of aerosols. However,

there is not yet a common understanding of the relevance of this nucleation pathway nor any measurements of such contrails. In this study, we have thus used the simplified modification of Gierens (2021), shown in Eq. (4). Here, $\bar{c}_p$ is a mol-based mean heat capacity of the exhaust gases (see Eq. (15) of Gierens, 2021), which is not a constant, but we nevertheless define to be 30.6 J/mol/K for all pressure levels for simplicity. This is possible because the variability of the heat capacity is low. $\eta_E$ the electric efficiency, $\eta_0$ is the basic efficiency and $\Delta h_m$ the formation enthalpy of water vapour. For simplicity we combine the

two efficiencies into a single efficiency $\eta$, as in Table 1.



$$G_{H_2FC} = \frac{\bar{c}_p p_a}{(1 - \eta_E \eta_0)|\Delta h_m|} \tag{4}$$

For any given mixing line slope $G$, we can define two threshold temperatures $T_{min}$ and $T_{max}$ to aid in determining where persistent contrails form. Consider the mixing line shown in blue in Figure 1 for $G = 3.7$ Pa/K, which is tangential to the water vapour saturation curve. $T_{max}$, sometimes also written $\Theta$ (e.g. Kärcher et al., 2015), refers to the temperature at which this threshold mixing line touches the water vapour saturation curve, in this case around 240.1 K; $T_{min}$ refers to the temperature at which the mixing line reaches a water vapour partial pressure of 0 Pa, in this case around 229.9 K. Both values increase with increasing $G$. The original approximation of $T_{max}$ by Schumann (1996) was extended by Gierens (2021) for higher temperatures (233 to 293 K) and higher mixing line slopes ($G > 2$ Pa/K),

$$T_{max} = \begin{cases} 226.69 + 9.43\ln{(G - 0.053)} + 0.72\left(\ln{(G - 0.053)}\right)^2 & \text{for } T < 233\ K \text{ or } G \leq 2\ Pa/K \\ 226.03 + 10.22\ln{G} + 0.335\left(\ln{G}\right)^2 + 0.0642\left(\ln{G}\right)^3 & \text{for } G > 2\ Pa/K \end{cases} \tag{5}$$

To determine whether a persistent contrail forms at a given location, we check the following three conditions, which we define as *limiting factors*:

1) *droplet formation* - ambient temperature less than the threshold temperature ($T < T_{max}$) and ambient water vapour partial pressure above the threshold mixing line;

2) *droplet freezing* - ambient temperature less than 235 K;

3) *persistence* - ambient water vapour partial pressure above the ice saturation curve.

If these three conditions are all met, a persistent contrail forms. For completeness, a fourth boundary, water supersaturation, can also be defined. However, for the pressure levels of interest in this article, the occurrence of these conditions is insignificant (Krämer et al., 2009). The limiting factors are shown graphically in Figure 1.

We note that in reality, the boundary definitions are less stringent. Previous studies have shown that nvPM activation and thus ice crystal number reduces as the ambient temperature approaches $T_{max}$ (Bräuer et al., 2021a) as well as 235 K (Bier et al., 2022, 2024; Kärcher, 2018). In the latter conditions, there is insufficient time in the plume for ice crystals to form before the water droplets evaporate. The effective freezing temperature limit is thus slightly colder and depends on the local relative humidity. This dependence is the subject of ongoing analysis and is thus not included here.

## 2.2 Aircraft definitions

For the limiting factors analysis, we define a range of aircraft, representative of different propulsion technologies. This is necessary to limit computational resources, but nevertheless allows for a good comparison between the technologies. We selected eight aircraft-fuel combinations that span a range of $G$ from 0.48 to 15.8 Pa/K at 250 hPa (Table 1).

The reference aircraft CON-LG corresponds to the last generation of conventional, kerosene-powered aircraft, for which we assume an overall propulsion system efficiency $\eta$ of 0.3. The next generation of kerosene-powered aircraft CON-NG has



an improved overall propulsion system efficiency of 0.4, therefore a colder exhaust and a higher $G$. We also define an 80 %
parallel hybrid-electric aircraft HYB-80. We use a high degree of electrification to differentiate it from the conventional aircraft
- a 60 % hybrid aircraft would have approximately the same mixing line slope as CON-LG. We further include two potential
aircraft designs that reduce the effective emission index of water vapour $EI_{H_2O}$ using the Water-Enhanced Turbofan (WET)
concept. In this study, we assume for simplicity a reduction in $EI_{H_2O}$ of 50 % (WET-50) and 75 % (WET-75). We ignore the

160 impact of any potential liquid droplets in the plume. We also define three different hydrogen-powered aircraft: H2C-04 is a
hydrogen combustion (H$_2$C) aircraft, which assumes $\eta = 0.4$; H2FC-LV and H2FC-HV are hydrogen fuel cell (H$_2$FC) aircraft
which operate at 0.5 and 1.0 V respectively. The operating voltage has a very large influence on the slope of the mixing line,
as Gierens (2021) showed. However, we only consider H2C-04 in this analysis, as explained hereafter.

**Table 1.** Definition of aircraft-engine-fuel combinations used in this study, representative of different current and future propulsion technologies. The aircraft IDs in brackets are not further considered in this study: H2C-04 is used as a proxy for all hydrogen-powered aircraft in this study since all have $G$ higher than the limit case of 4.29 Pa/K, as described in the main text.

| Aircraft ID | Description | Eq. | $c_p$ | $EI_{H_2O}$ | Q or $\Delta$h | $\eta$ | G (250 hPa) |
|---|---|---|---|---|---|---|---|
| WET-75 | -75% EI$_{H_2O}$ | (1) | 1004 J/kg/K | 0.31 kg/kg | 43.6e6 J/kg | 0.4 | 0.48 |
| WET-50 | -50% EI$_{H_2O}$ | (1) | 1004 J/kg/K | 0.63 kg/kg | 43.6e6 J/kg | 0.4 | 0.97 |
| HYB-80 | Hybrid (80%) | (2) | 1004 J/kg/K | 1.25 kg/kg | 43.6e6 J/kg | 0.4 (K), 0.8 (E) | 1.15 |
| CON-LG | Last gen. JA1 | (1) | 1004 J/kg/K | 1.25 kg/kg | 43.6e6 J/kg | 0.3 | 1.65 |
| CON-NG | Next gen. JA1 | (1) | 1004 J/kg/K | 1.25 kg/kg | 43.6e6 J/kg | 0.4 | 1.93 |
| H2C-04 | H$_2$C | (3) | 1004 J/kg/K | 8.94 kg/kg | 120.9e6 J/kg | 0.4 | 4.97 |
| (H2FC-LV) | H$_2$FC $\sim$0.5V | (4) | 30.6 J/mol/K | - | -241.82 kJ/mol | 0.4 | 5.27 |
| (H2FC-HV) | H$_2$FC $\sim$1.0V | (4) | 30.6 J/mol/K | - | -241.82 kJ/mol | 0.8 | 15.8 |

With the definition of the homogeneous freezing temperature at 235 K, we can calculate a mixing line slope that is tangential

to the water saturation curve and crosses the intersection of the ice saturation curve with the homogeneous freezing temperature.
Using the Newton method, we find this slope to be $G_{lim} = 4.29$ Pa/K. This mixing line slope is the theoretical limit case: For
$G > 4.29$ Pa/K, whether a contrail forms or not is no longer aircraft design dependent, but purely depends on the ambient
conditions. H2C-04 has a mixing line slope lower than 4.29 Pa/K only at altitudes higher than 217 hPa; H2FC-LV higher
than 205 hPa; and H2FC-HV always has a mixing line slope greater than 4.29 Pa/K. Since it is very unlikely that fuel cell

aircraft will fly at these altitudes and, as we show in our results (cf. Figure 5), persistent contrail formation is not aircraft design
dependent at high altitudes, in this study H2C-04 is used as a proxy for all hydrogen-powered aircraft.

### 2.3 Air traffic data

We use a representative future air traffic scenario to weight our results and identify areas of particular interest. In this study, we
use the progressive scenario of the DLR internal project *Development Pathways for Aviation up to 2050* (DEPA 2050, Leipold

et al., 2021). A key objective of this project was to define and assess the development of the aviation industry until the year



2050 from multiple angles, including climate, economics and society. It also studied the potential entry into service of various novel aviation fuels and technologies. This scenario was selected for this research since it is representative of a future multi-fuel global fleet and because there are no restrictions on its publication. The DEPA 2050 progressive scenario for the year 2050 is included in the linked data and is shown graphically in Supplementary Figure 1.

## 2.4 ERA5 data and corrections

To calculate regions of potential persistent contrail formation we use the European Centre for Medium-Range Weather Forecasts (ECMWF) ERA5 reanalysis data (Hersbach et al., 2020) stored on the German Climate Computing Center's (DKRZ) supercomputer Levante. Specifically, we use the ambient temperature and relative humidity at pressure levels 350, 300, 250, 225, 200, 175 and 150 hPa within the 2010 decade (December 2009 to November 2019 inclusive). In this study, we use RHi to denote relative humidity with respect to ice and RHw to denote relative humidity with respect to water. We further define the seasons Winter, Spring, Summer and Autumn as DJF, MAM, JJA, and SON respectively. To avoid autocorrelation and reduce the computational time, whilst still obtaining a climatological response, we randomly select 2160 hours per season over the 10-year time frame - this corresponds to 10 % of all hourly values within the 2010 decade. We use *numpy.random.choice* with seed 42 to determine which hours to choose per season for repeatability. Supplementary Figure 2 shows the hours that were selected.

The ERA5 HRES temperature and relative humidity values are archived on a reduced Gaussian grid with a resolution of N320. The pre-interpolated data is available on DKRZ Levante as daily GRIB files. We decided to use the irregular N320 grid for our analysis, rather than interpolated data, to avoid any artefacts of interpolation in the relative humidity data in particular. However, this complicates the definition of cell neighbours and the calculation of the limiting factors, as described in the next section.

Numerous authors have shown the difficulty of estimating ice supersaturated regions (ISSRs) and hence contrail persistence with reanalysis data. For example, Agarwal et al. (2022) found that ERA5 could overestimate ISSR coverage when compared to radiosonde measurements, whereas Reutter et al. (2020) found an underestimation in comparison to in situ IAGOS measurements (Petzold et al., 2015). Gierens et al. (2020), comparing ERA5 with MOZAIC data (Marenco et al., 1998), found that although contrail formation can be predicted reliably, persistence cannot. In a follow-up study, Hofer et al. (2024b) found that even using further ERA5 variables, ISSR prediction cannot yet be significantly improved.

There have been many suggestions on how to correct ERA5 data to more accurately estimate ISSRs, both at a regional (Wang et al., 2024; Teoh et al., 2022a) and global scale (Wolf et al., 2023b; Hofer et al., 2024b; Teoh et al., 2024). In this study, we enhance the ERA5 humidity by applying a factor of $1/\text{RHi}_\text{C}$ for $\text{RHi}_\text{C} \in \{1.0, 0.98, 0.95, 0.90\}$, as in previous studies (e.g. Schumann, 2012; Teoh et al., 2020; Schumann et al., 2021; Reutter et al., 2020). We note that this approach does not consider a latitude-dependence of the ERA5 RHi error (Teoh et al., 2024). However, modifying the cumulative distribution function found for ERA5 by Hofer et al. (2024b, their Figure 4) to include the RHi enhancements, we find that $\text{RHi}_\text{C} = 0.95$ provides a good fit against MOZAIC/IAGOS data for $\text{RHi} \leq 1.0$ (Supplementary Figure 3), which is of relevance for this study. We further discuss the regional implications of this method in Section 3.6.



## 2.5 Limiting factors analysis

The objective of the limiting factors analysis is to understand the aircraft design-, altitude-, latitude- and seasonal dependence of the factors responsible for persistent contrail formation. This helps us identify regions where the introduction of aircraft using novel propulsion systems and fuels with substantially higher mixing line slopes, e.g. hydrogen combustion, would lead to increased persistent contrail formation. Or, where the use of aircraft with substantially lower mixing line slopes, such as hybrid-electric or WET aircraft, could be particularly beneficial.

We first consider the probability of persistent contrail formation $p_{pcf}$ for each aircraft design. This is calculated by creating a boolean mask for persistent contrail formation using the three limiting factors for all 3D grid cells, time steps and aircraft designs. The boolean values for each grid cell are averaged and saved per month. Since the underlying ERA5 grid is irregular, when reporting and analysing a global $p_{pcf}$, we use an area- and level-weighting. We calculate the area of all grid cells as viewed from above using the *calculate_areas* method of the Python library *scipy.spatial.SphericalVoronoi* with an assumed spherical Earth radius of 6371 km and assume the cell areas to be constant with altitude. Since ERA5 also has irregular spacing in altitude, we use the difference (in hPa) between pressure levels to level-weight the results.

We then consider each limiting factor individually. For each aircraft design, time step and grid cell, we determine which of the limiting factors prevent a persistent contrail from forming. By summing the number of times each factor was limiting and normalising by the number of time steps, we obtain the probability of each limiting factor preventing persistent contrail formation. The results are saved per season-year (e.g. 2010 DJF) to limit memory usage. We can then investigate the dependence of each limiting factor on altitude, latitude and season. When reporting global results, we perform area- and level-weighting as before.

## 2.6 Analysis of persistent contrail formation region boundaries

We now consider again Figure 1: If an aircraft is flying with the mixing line slope $G$ shown by the blue line in ambient conditions within the green hatched area, a persistent contrail will form. If the ambient conditions are subsequently no longer within the green hatched area, due to changes in the ambient temperature, relative humidity, altitude or in aircraft performance, the aircraft will stop producing a persistent contrail, thus defining the boundary of a persistent contrail formation region. One or more of the limiting factors will have been responsible for that boundary.

The objective of this analysis is to understand the relative importance of the limiting factors in defining the horizontal and vertical boundaries of persistent contrail formation regions. In horizontal direction, we begin by calculating the total perimeter of all persistent contrail formation regions. Then, we compare the length of the boundary caused by each limiting factor to the total perimeter. The boundary lengths for each limiting factor are thus subsets of the total. But since multiple factors can be simultaneously limiting, the sum of all limiting factor boundary lengths is greater than the total. A large boundary length for a given limiting factor means that that factor was often responsible for persistent contrails forming.

We demonstrate this analysis graphically in an example shown in Figure 2 for a single time and pressure level over the Netherlands. For a given aircraft design, green cloud icon nodes represent grid points where a persistent contrail would form;



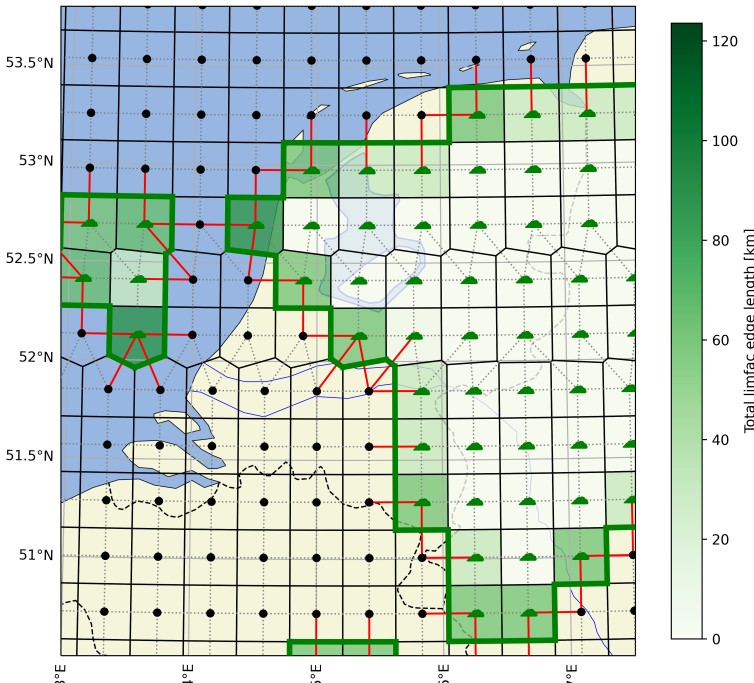

**Figure 2.** Example of the horizontal total limiting factors analysis for a hydrogen combustion aircraft (H2C-04) at a single time (February 11th, 2024 at 08:00 UTC) at a single level (300 hPa) over the Netherlands as calculated with ERA5 without RHi correction. Persistent contrails would form for this aircraft at all grid points with a green cloud icon. The thick green lines correspond to the edge of the potential persistent contrail formation regions; the red lines to all possible boundary crossings; and the shading to the length of the persistent contrail formation region boundary around each grid cell.

black points where none are expected. We calculate the borders between grid cells (solid black lines) using the *scipy.spatial.SphericalVoronoi* method, and find cell neighbours (gray dotted lines) by determining which cells share a pair of edge vertices.

We are interested in the boundaries of persistent contrail formation regions, therefore we need to identify neighbouring cells with different icons (red, solid lines). These cells $i$ and $j$ share a border $d_{ij}$ (thick green line), which we calculate using the Haversine formula assuming an Earth radius of 6371 km. The colour gradient bar indicates the length of the persistent contrail formation region border around each grid cell: a darker green corresponds to a longer boundary, a lighter green to a shorter boundary. By globally summing all $d_{ij}$, we obtain the total perimeter of all persistent contrail formation regions. This

is repeated for the boundaries caused by individual limiting factors.

Programmatically, for each aircraft design and time step, we create four 3D boolean masks: one for persistent contrail formation and one per limiting factor. For each aircraft design, time step, pressure level and limiting factor, we create a global,





symmetric adjacency matrix $A$ of size $n \times n$, where $n = 542080$ is the number of grid points in the ERA5 grid. For any two adjacent cells $i$ and $j$, $A_{ij} = d_{ij}$ (shared border length, thick green line) if the persistent contrail formation boolean at $i$ is TRUE

(persistent contrail forms, green cloud icon) and the limiting factor boolean at $j$ is FALSE (the limiting factor has prevented persistent contrail formation, black point), else $A_{ij} = 0$. To obtain the total perimeter, we set $A_{ij} = d_{ij}$ if *any* limiting factor boolean at $j$ is FALSE. We then obtain the vector of size $n$ of boundary lengths around each grid cell (cell colour gradient) by summing the columns of the adjacency matrix. This procedure is repeated for each time step and the monthly summed values for each aircraft design, grid cell, pressure level and limiting factor are saved to file. We repeat the full analysis for all RHi

enhancements.

We perform a similar analysis in the vertical direction to analyse the impact of changing altitude and thus mixing line slope $G$. For simplicity, we neglect the changes in aircraft operating conditions during climb or descent to a new altitude. Vertically neighbouring cells can be found easily because the latitude-longitude location of the cells is constant with altitude. Since there are no clearly defined edge lengths in the vertical direction, we calculate the boundaries of persistent contrail formation regions

by summing the cell areas as viewed from above.

### 2.7 Maximum slope analysis

In the limiting factors analysis, fixed aircraft designs with specific values of $G$ are used. The objective of this second analysis is to explore the dependence of persistent contrail formation $p_{pcf}$ on $G$ in more detail, and determine whether there is a climatological relationship between the two. We analyse the results at the global level as well as on three latitude-dependent

scales: the northern extratropics (latitude > 30°N), the tropics (30°S ≤ latitude ≤ 30°N) and the southern extratropics (latitude < 30°S).

For each cell where the ambient conditions are sufficient for persistent contrail formation, we use the Newton method to find the slope where the mixing line is tangent to the water saturation curve. The slope of this mixing line we define as $G_{max}$ - the maximum mixing line slope for which a persistent contrail would not form. For example, in Figure 1 the blue line corresponds

to $G_{max} = 3.7$ Pa/K. For each region, altitude and time step, we use a histogram with a bin size of 0.2 Pa/K to determine the distribution of $G_{max}$. By taking the cumulative sum of the histograms, we obtain $p_{pcf}$ as a function of $G$.

## 3 Results and Discussion

This section presents and discusses the results of the limiting factors analyses. Sections 3.1 and 3.2 analyse the limiting factors within each grid cell and at the boundaries of persistent contrail formation regions. The altitude- and latitude-dependence of

these regions and the changes in global potential persistent contrail formation $p_{pcf}$ are presented in Section 3.3. In Section 3.4, we develop climatological relationships between the mixing line slope and the $p_{pcf}$. Finally, we discuss the interplay and competition between the limiting factors in Section 3.5 and the limitations and uncertainties of our study in Section 3.6.



### 3.1 Limiting factors of persistent contrail formation

We find that the formation of persistent contrails is primarily limited by ice supersaturation (persistence limiting factor).

Figure 3 shows the global area- and level-weighted potential persistent contrail formation ($p_{pcf}$) and the frequency that each limiting factor prevented a persistent contrail from forming. Enhancing the area of ice supersaturation by lowering the relative humidity threshold as a means of correcting ERA5 relative humidity (see Sect. 2.4) increases $p_{pcf}$, proportionally more for aircraft with higher $G$ (markers).

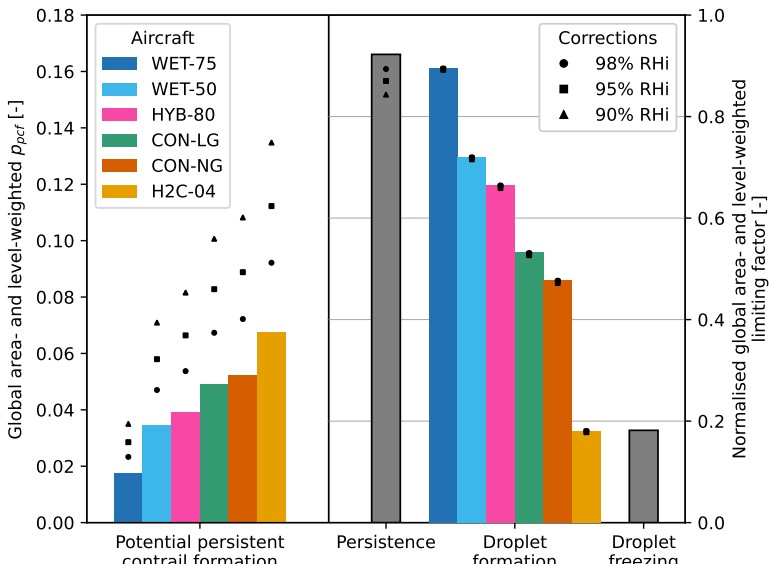

**Figure 3.** Global area- and level-weighted potential persistent contrail formation during the 2010 decade (left) and the normalised frequency that each limiting factor prevented a persistent contrail from forming (right). A normalised frequency of 0 would mean that the factor never limited persistent contrail formation; 1 that it always does. Dependence on aircraft design is shown by the different colours - the persistence and droplet freezing limiting factors are independent of aircraft design and thus only have single values. The RHi enhancements are shown as scatter points - note the difference in the y-scale.

The persistence limiting factor is less limiting in the extratropics at lower altitudes (350 to 250 hPa) and in the tropics at
higher altitudes (225 to 150 hPa, see Supplementary Figure 4), but in total prevents persistent contrail formation 92.3 % of the time. Enhancing humidity by lowering the threshold for ice supersaturation reduces how limiting persistence is, as expected. On the other hand, droplet freezing is generally only limiting in the tropics at altitudes below 300 hPa, in total only 18.2 % of the time. It is by definition unaffected by humidity enhancements.

Droplet formation is highly dependent on the slope of the mixing line $G$ and varies between 18.1 % (H2C-04) and 89.5 %
(WET-75). It varies insignificantly with enhanced humidity. An interesting finding is that the H2C-04 droplet formation and the droplet freezing factors are very similar, both in their normalised sum as in their latitude- and altitude-dependence (Supple-





mentary Figure 4). The reason for this is that there is some overlap in the definition of the limiting factors: Ambient conditions that are to the right of the steep mixing line will likely also be to the right of the droplet freezing line. The result is that the droplet formation limiting factor acts like persistence at low $G$ and like droplet freezing at high $G$.

## 3.2 Boundaries of persistent contrail formation regions

Persistence is also found to be more limiting compared to the other factors at the region boundaries. Figure 4 shows the sum over all pressure levels of the horizontal (a) and vertical (b) boundary lengths/areas of potential persistent contrail formation regions as a function of the limiting factor and aircraft design. We show the total in absolute terms and the individual limiting factors relative to the total for ease of comparison. Because more than one limiting factor may simultaneously be responsible

for a boundary, the sum of the individual limiting factors may be larger than 100 %. Since the horizontal and vertical directions cannot be directly compared, we have aligned the total limiting factor for CON-LG such that at least the shapes of the responses can be compared. In both cases, the persistence limiting factor dominates for all aircraft designs and increases with increasing $G$. This is because as $G$ increases, contrails can form in larger proportions of ice supersaturated regions.

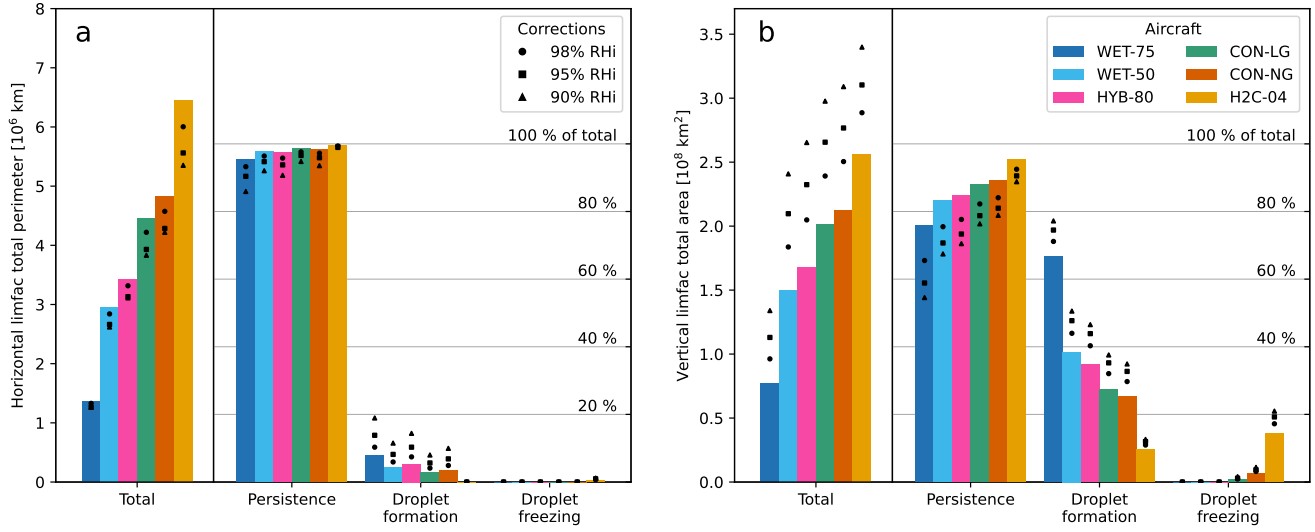

**Figure 4.** Horizontal **(a)** and vertical **(b)** total limiting factor results. The "total" limiting factors are shown in absolute values, visually aligned for CON-LG such that the shapes of the responses can be compared; the individual limiting factors are shown relative to the corresponding total value and are given in percent. Because more than one factor may simultaneously be limiting, the sum of the individual percentages may be larger than 1. The RHi enhancements are shown as scatter points.

Droplet formation can be limiting when the ambient conditions would be sufficient for persistent contrail formation, but $G$

is sufficiently low that the exhaust does not become supersaturated with respect to water. The droplet formation limiting factor thus becomes less relevant as $G$ increases. In horizontal direction, it is no longer relevant once $G > 4.29$ Pa/K since for all



ambient conditions conducive to persistent contrail formation the mixing line would cross the water saturation curve. Since $G$ changes with altitude for the same aircraft, in the vertical direction droplet formation is responsible for a significantly larger proportion of the persistent contrail formation region boundaries. It is particularly limiting for very low $G$ aircraft such as the

WET-75.

In comparison, the droplet freezing limiting factor becomes more important with increasing $G$. Low mixing line slopes prevent contrails from forming near the freezing boundary. Therefore, droplet freezing is only relevant for aircraft with a high $G$, in particular hydrogen-powered aircraft. Again due to the changing $G$ with altitude and higher change in ambient temperature in vertical than in horizontal direction, the droplet freezing limiting factor is also more relevant in the vertical

direction.

The RHi correction does not influence which factor is most limiting, with the single exception of WET-75 in vertical direction. It also does not, on a climatological timescale, influence in which areas persistent contrails form: As the humidity is enhanced, the probability that a persistent contrail forms increases where the probability was already greater than zero (cf. Supplementary Figure 5). However, the persistence results show that many ambient conditions in ERA5 are close to being

saturated with respect to ice. Our results thus only add to the consensus that improving ISSR forecasting is vital for estimating persistent contrail formation (Gierens et al., 2020; Hofer et al., 2024b).

In horizontal direction, enhancing humidity seems to reduce the total edge length, even though the potential persistent contrail formation regions are larger. A significant reduction in edge length can be identified by enhancing humidity by only 2 %, in particular for hydrogen-powered aircraft. This suggests that the potential persistent contrail formation regions, predominantly

described by ice supersaturation, grow and coalesce with enhancing humidity, reducing the total length of their boundaries. In horizontal direction, the RHi correction has only a minor influence on the droplet freezing, but a major influence on the droplet formation limiting factor. In both cases, however, the total perimeter remains two orders of magnitude smaller than that of the persistence limiting factor. In vertical direction, enhancing humidity increases the boundary area significantly for the total. Seemingly, the boundary between different pressure levels is emphasised by the enhanced humidity, notably for the droplet

formation limiting factor.

### 3.3 Spatial variability of persistent contrail formation

The summation of the region edges does not provide any information about where persistent contrails form. Therefore, we now explore the altitude-, latitude- and aircraft-dependence of persistent contrail formation regions. Figure 5(a) shows that aircraft design is only ever limiting at lower altitudes: Except for aircraft with very low mixing line slopes (HYB-80, WET), whether

a contrail forms and persists above 200 hPa depends solely on the ambient conditions.

It has been widely assumed that the introduction of hydrogen-powered aircraft would result in a significant increase in the formation of persistent contrails. Our results show that a hydrogen-powered aircraft, represented by H2C-04, could indeed form many more persistent contrails than existing conventional aircraft (CON-LG). Comparing the two aircraft designs directly, our results match those of Kaufmann et al. (2024, the right side of their Figure 5): At low altitude (350 to 300 hPa), a large difference

in persistent contrail formation (Kaufmann et al. (2024) call this "potential contrail cirrus cover") can be seen between 30 to





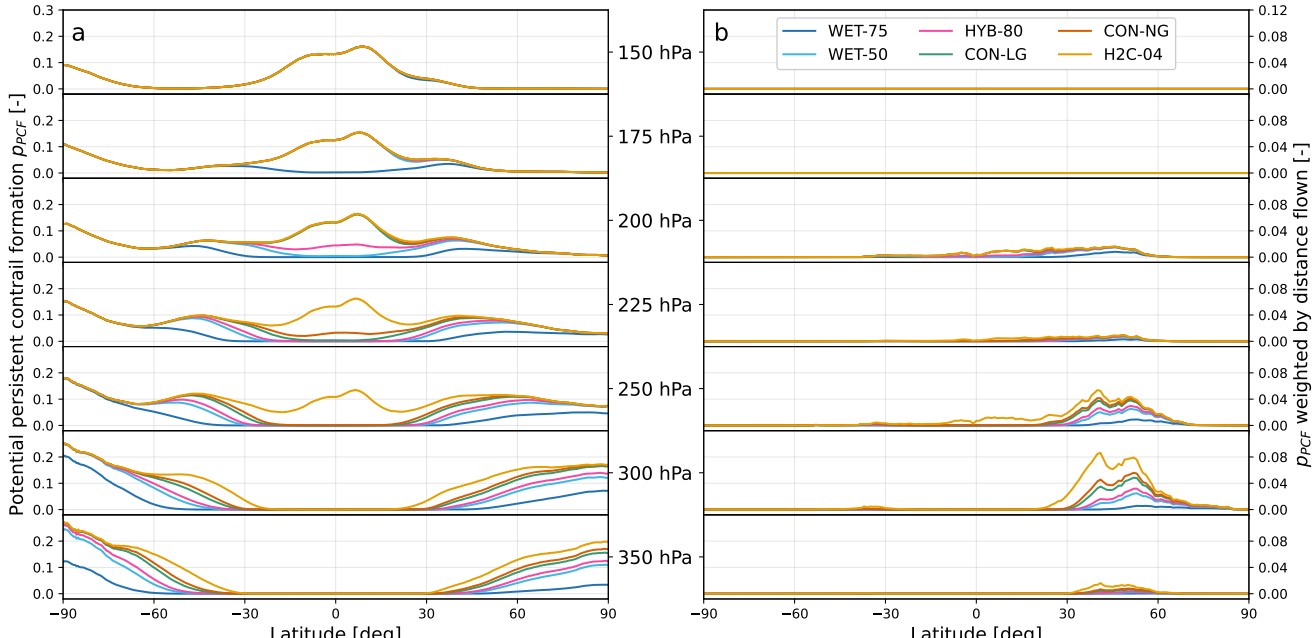

**Figure 5.** Potential persistent contrail formation $p_{pcf}$ **(a)** as a function of latitude (x-axis) and altitude (secondary y-axis). $p_{pcf}$ is then weighted by the distance flown in the DEPA 2050 air traffic scenario in the year 2050 **(b)** to identify important areas for persistent contrail formation.

60°S and 30 to 60°N, coinciding with significant air traffic over Europe and the US. However, the largest difference occurs in the tropics between 250 and 225 hPa: 17.9 and 5.2 times more persistent contrails could be formed respectively (Supplementary Table 1). In total, across all latitudes and considered altitudes, hydrogen-powered aircraft have the potential to produce 25.7 % more persistent contrails than conventional aircraft (Table 2).

Since these results do not consider where aircraft actually fly, we have also used the DEPA 2050 air traffic scenario to weight the results, as shown in Figure 5(b). DEPA 2050 is only available with a 1° latitude resolution, so we conservatively regrid the limiting factors results to match this resolution. The tropical increase in persistent contrail formation at 225 to 250 hPa is now less pronounced because there are few aircraft flying in this region. Instead, of particular importance are the northern extratropics between 300 and 250 hPa (FL300 to FL340), which are the lower cruise altitudes for typical airliners. In total, our

simple distance-weighted proxy shows a globally-averaged increase of 71.4 % (Table 2) for hydrogen-powered compared to conventional aircraft. This indicates that replacing existing aircraft along existing and expected flight routes with hydrogen-powered designs would significantly increase persistent contrail formation - on a climatological timescale whilst ignoring any real-time avoidance strategies.

Significant reductions in persistent contrail formation could be gained by reducing the water vapour emission index: globally,

WET-50 achieves a 23.3 % reduction; WET-75 a 55.1 % reduction. Along the DEPA 2050 flight routes, this further improves





**Table 2.** Potential persistent contrail formation $p_{pcf}$ and DEPA 2050 distance-weighted $p_{pcf}$ for the pressure levels 300 and 250 hPa and all considered pressure levels, averaged over four latitude bands: the southern extratropics (xtropS), tropics (trop), nothern extratropics (xtropN) and all latitudes. Percentage changes are provided for the globally-averaged values for simple comparison. The full data is available within the linked dataset.

| Aircraft | Pressure level | $p_{pcf}$ | | | | Distance-weighted $p_{pcf}$ | | | |
|---|---|---|---|---|---|---|---|---|---|
| | | xtropS | trop | xtropN | all | xtropS | trop | xtropN | all |
| WET-75 | 300 hPa | 0.057 | 0.000 | 0.028 | 0.028 (38.9 %) | 0.000 | 0.000 | 0.002 | 0.001 (11.5 %) |
| | 250 hPa | 0.061 | 0.000 | 0.030 | 0.030 (47.2 %) | 0.000 | 0.000 | 0.003 | 0.001 (17.4 %) |
| | all | 0.044 | 0.015 | 0.019 | 0.026 (44.9 %) | 0.000 | 0.000 | 0.001 | 0.000 (16.3 %) |
| WET-50 | 300 hPa | 0.096 | 0.000 | 0.064 | 0.053 (73.0 %) | 0.000 | 0.000 | 0.008 | 0.003 (46.2 %) |
| | 250 hPa | 0.086 | 0.000 | 0.064 | 0.050 (77.5 %) | 0.000 | 0.000 | 0.009 | 0.003 (57.3 %) |
| | all | 0.064 | 0.029 | 0.040 | 0.044 (76.6 %) | 0.000 | 0.000 | 0.004 | 0.001 (52.8 %) |
| HYB-80 | 300 hPa | 0.104 | 0.000 | 0.075 | 0.060 (82.1 %) | 0.000 | 0.000 | 0.011 | 0.004 (61.1 %) |
| | 250 hPa | 0.093 | 0.000 | 0.073 | 0.055 (85.7 %) | 0.000 | 0.000 | 0.011 | 0.004 (71.6 %) |
| | all | 0.069 | 0.033 | 0.045 | 0.049 (84.6 %) | 0.000 | 0.000 | 0.004 | 0.002 (67.4 %) |
| CON-LG | 300 hPa | 0.119 | 0.000 | 0.100 | 0.073 (100.0 %) | 0.000 | 0.000 | 0.017 | 0.006 (100.0 %) |
| | 250 hPa | 0.104 | 0.004 | 0.086 | 0.065 (100.0 %) | 0.000 | 0.000 | 0.015 | 0.005 (100.0 %) |
| | all | 0.076 | 0.043 | 0.055 | 0.058 (100.0 %) | 0.000 | 0.001 | 0.006 | 0.002 (100.0 %) |
| CON-NG | 300 hPa | 0.126 | 0.000 | 0.108 | 0.078 (107.1 %) | 0.000 | 0.000 | 0.021 | 0.007 (120.3 %) |
| | 250 hPa | 0.107 | 0.008 | 0.090 | 0.068 (105.5 %) | 0.000 | 0.001 | 0.016 | 0.006 (111.6 %) |
| | all | 0.079 | 0.047 | 0.058 | 0.061 (105.5 %) | 0.000 | 0.001 | 0.007 | 0.003 (113.8 %) |
| H2C-04 | 300 hPa | 0.145 | 0.004 | 0.132 | 0.094 (129.4 %) | 0.001 | 0.002 | 0.034 | 0.012 (206.0 %) |
| | 250 hPa | 0.111 | 0.080 | 0.099 | 0.097 (149.4 %) | 0.001 | 0.008 | 0.019 | 0.009 (171.6 %) |
| | all | 0.085 | 0.067 | 0.067 | 0.073 (125.7 %) | 0.000 | 0.003 | 0.010 | 0.004 (171.4 %) |

to 47.2 % and 83.7 % respectively. There is also clearly potential for the targeted introduction of aircraft with very low mixing line slopes, in particular at higher altitudes (225 to 200 hPa, FL360 - FL380) in the tropics. Here, the WET aircraft considered would produce little to no persistent contrails at all, with unweighted reductions of 83 - 87 % (WET-50) and 99.5 - 99.95 % (WET-75).

For all aircraft designs, persistent contrail formation increases with altitude in the tropics and decreases in the extratropics, in line with previous studies (e.g. Matthes et al., 2021). Among others, Barton et al. (2023) suggest designing hydrogen-powered aircraft capable of flying at higher altitudes in the extratropics - around 14 km (140 hPa) - to avoid forming persistent contrails. We find that very few persistent contrails are formed at 150 hPa above 30°N and between -30°S and -60°S (very few flight tracks extend below -60°S). Since this is due to a lack of ice supersaturation in these regions (see Supplementary Figure 4),

this concept does not need to be limited to hydrogen-powered aircraft. However, these altitudes are above the local tropopause, and releasing significant amounts of water vapour and $NO_x$ into the stratosphere can result in a significant increase in climate



impact (Matthes et al., 2021; Grewe et al., 2010). We therefore stress that these results should not be used in isolation to determine a possible optimum cruise altitude. Rather, the full climate impact of a given design must be considered.

### 3.4 Climatological potential persistent contrail formation

We now wish to determine whether there is a climatological relationship between the mixing line slope $G$ and the potential persistent contrail formation $p_{pcf}$. Figure 6 shows $p_{pcf}$ as a function of the mixing line slope $G$ for each pressure level and for all data together, without RHi enhancement. The results for all RHi enhancements are shown in Supplementary Figure 6. An asymptote in the data at $G_{lim} = 4.29$ Pa/K exists by definition (see Section 2.2), but for many pressure levels 99% of that maximum value is actually achieved at much lower mixing line slopes ($G_{99\%sup}$ in Table 3 and black circular markers in 380 Figure 6). Generally, the higher the altitude, the earlier the asymptote and the less influence the aircraft design has. This also matches the results found in Figure 5.

For very low $G$, persistent contrails form only at very high altitude. As $G$ increases, more persistent contrails are formed at lower altitudes. The altitude with the highest $p_{pcf}$ is likely between 250 and 225 hPa at $G > 4$ Pa/K. For these two altitudes in particular, the results seem to be the addition of two different responses. We find that the initial slope is the extratropical 385 response (both northern and southern), and the second slope the tropical response (right side of Figure 6). Given the higher tropopause and ambient temperature in the tropics, persistent contrails generally form at higher values of $G$ compared to the extratropics.

We fit the responses using the sum of two logistic functions, one representing the response from the tropics and the other from the extratropics. A conventional logistic function includes the supremum $L$, the growth rate $k$ and the midpoint of the 390 growth $x_0$, with limits 0 for $x \to -\infty$ and $L$ for $x \to +\infty$. We have allowed the extratropical logistic function to be shifted vertically by $d$ for a better fit, noting that the fitting function is only valid for $G > 0$ Pa/K. To find the optimal parameters, we consider the tropical logistic function to be an addition to the extratropical function. Therefore, we split the data into two at a certain value of $G$ and use the values of $G$ up until this split to create the extratropical response, and the values after the split to create the tropical response. We subsequently use a minimisation function to determine the best value for the split.

$$f(x) = \frac{L_1}{1 + e^{-k_1(x - x_{0,1})}} + \frac{L_2}{1 + e^{-k_2(x - x_{0,2})}} + d \qquad (6)$$

The result is that the value of the maximum for the full equation can be read as $L_1 + L_2 + d$. For the altitudes 175 and 150 hPa, we find that there is no need to use a second logistic function because persistent contrails do not form at these altitudes in the extratropics. For these altitudes, therefore, the value of the asymptote is simply $L + d$. Since there is by definition an asymptote at $G_{lim} = 4.29$ Pa/K, we only grouped values of $G$ up until $G = 4.5$ Pa/K. To help the minimisation function describe the 400 asymptotic behaviour above this limit value, we artificially extended the results to 6.5 Pa/K. This is represented as the faded part of Figure 6.

The parameters found by the minimisation function are shown in Table 3. We generally find a good fit and a high coefficient of determination ($R^2$), except for 150 hPa and all level data together. At 150 hPa, the seasonal responses reach an asymptote at a



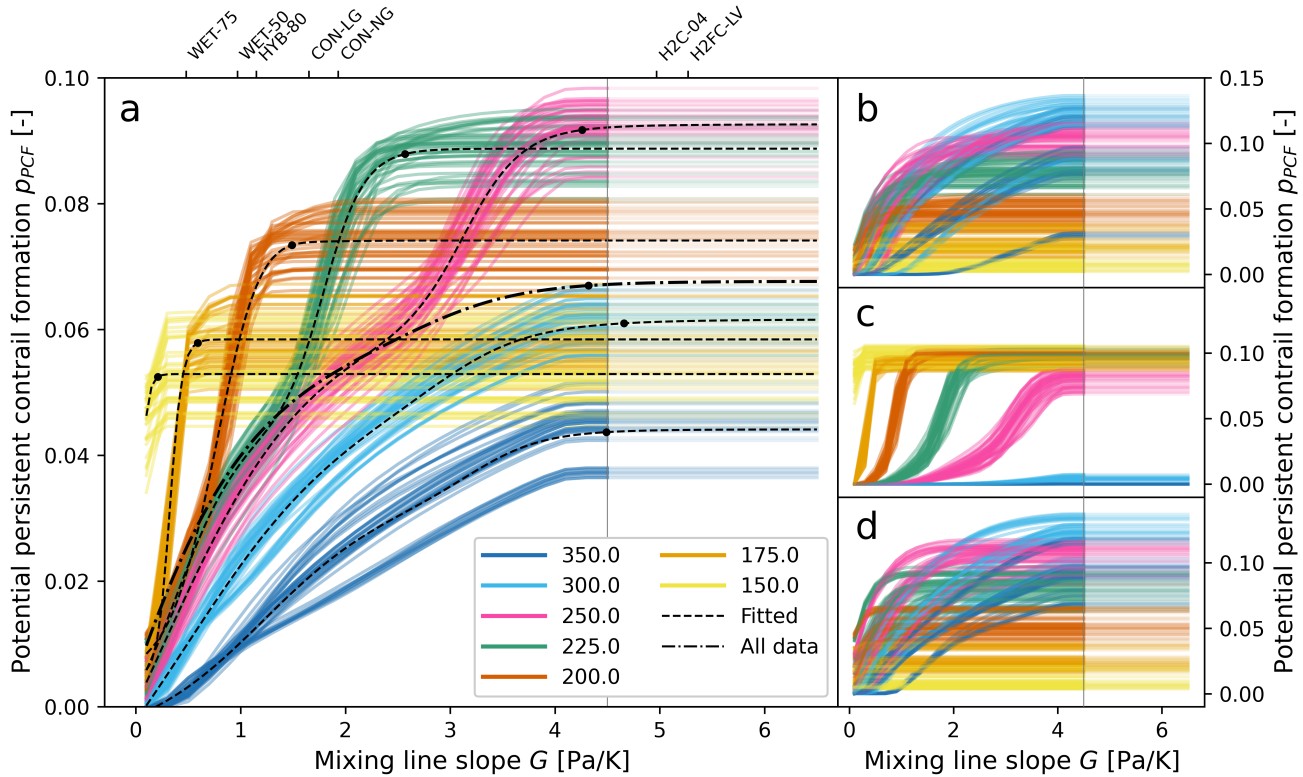

**Figure 6.** Potential persistent contrail formation $p_{pcf}$ as a function of the mixing line slope $G$ on a climatological timescale for pressure levels 350 to 150 hPa **(a)** at a global scale and for **(b)** the northern extratropics, **(c)** the tropics and **(d)** the southern extratropics. Each coloured line corresponds to a single season within the 2010 decade. The dashed lines are the fitted responses using the modified logistic function in Eq. (6) and parameters as defined in Table 3, for each pressure level individually ("fitted") and for all pressure levels together ("all data"). The faded part for $G > 4.29$ Pa/K represents where the data was extended to improve the accuracy of the fits (see main text for more details). The markers show at which value of $G$ the corresponding $p_{pcf}$ reaches within 1 % of the supremum. The mixing line slopes of the different aircraft designs are shown for the pressure level 250 hPa for reference.

very low $G$, such that the response is essentially horizontal and thus by definition cannot have a good $R^2$ value. By considering

all level data together, the differentiation between the extratropical and tropical responses are essentially no longer visible. The $R^2$ value is thus not high, clearly showing that including level information is beneficial in determining $p_{pcf}$. We note that the "all data" response in Figure 6 has a similar shape to the gentle sloping curve in Figure 2 of Hofer et al. (2024a), who consider persistent contrail formation from potential SAF aircraft using IAGOS/MOZAIC data, irrespective of the altitude.

As Figure 3 demonstrated, $p_{pcf}$ increases with enhanced relative humidity because more ambient conditions, in particular

closer to the freezing temperature limit, are conducive to persistent contrail formation. The same analysis as above is conducted for all RHi enhancements and shown in Supplementary Figure 6. The fitted results are stretched to higher $p_{pcf}$ but maintain



**Table 3.** Optimal parameters for potential persistent contrail formation $p_{pcf}$ as a function of the mixing line slope $G$ (Eq. 6) on a climatological timescale for pressure levels 350 to 150 hPa and for all pressure levels together ("all data"). $L + d$ corresponds to the maximum value, shown in percentage. For each pressure level, the top values of $L$, $k$ and $x_0$ are for the first logistic function (extratropical response) and the bottom values for the second logistic function (tropical response). The pressure levels 175 and 150 hPa are primarily driven by the tropical response (cf. Figure 6) and thus did not benefit from the second logistic function. $G_{99\%sup}$ corresponds to the lowest value of $G$ for which the corresponding $p_{pcf}$ is within 1 % of the supremum $L + d$.

| Level | L | k | $x_0$ | d | L+d [%] | $R^2$ | $G_{99\%sup}$ [Pa/K] |
|---|---|---|---|---|---|---|---|
| All data | 0.167 | 1.35 | -0.578 | -0.109 | 6.77 | 0.446 | 4.32 |
| | 0.010 | 2.00 | 2.78 | | | | |
| 350 hPa | 0.040 | 1.63 | 1.21 | -0.006 | 4.41 | 0.937 | 4.49 |
| | 0.010 | 2.99 | 3.24 | | | | |
| 300 hPa | 0.082 | 1.24 | 0.47 | -0.032 | 6.16 | 0.978 | 4.66 |
| | 0.011 | 2.49 | 2.97 | | | | |
| 250 hPa | 0.107 | 1.37 | 0.21 | -0.046 | 9.26 | 0.985 | 4.26 |
| | 0.031 | 3.70 | 3.14 | | | | |
| 225 hPa | 0.055 | 3.17 | 0.38 | -0.010 | 8.87 | 0.981 | 2.57 |
| | 0.044 | 5.15 | 1.79 | | | | |
| 200 hPa | 0.077 | 4.32 | 0.78 | 0.005 | 7.41 | 0.957 | 1.49 |
| | -0.007 | 5.16 | 1.38 | | | | |
| 175 hPa | 0.054 | 17.3 | 0.32 | 0.005 | 5.84 | 0.865 | 0.59 |
| 150 hPa | 0.318 | 24.6 | -0.05 | -0.266 | 5.29 | 0.050 | 0.21 |

the same general shape, including the marked difference in the extratropical and tropical responses. This again suggests that, on a climatological timescale, as relative humidity is enhanced, the probability that a persistent contrail forms increases where the probability was already greater than zero (cf. Supplementary Figure 5).

## 3.5 Competition between limiting factors

We find that for all aircraft designs, persistence is the most limiting factor and most responsible for the borders of persistent contrail formation regions (Figures 3 and 4 respectively). At first glance, this seems counterintuitive: Persistence is independent of aircraft design, and yet the size of the potential persistent contrail formation regions do depend on the aircraft design and vary significantly. In this section, we discuss the interplay and competition between the different limiting factors.

Consider the analysis of persistent contrail formation regions in horizontal direction for the H2C-04, a high-$G$ aircraft. Since most region boundaries are defined by persistence and we do not change the operating condition of the aircraft between grid cells, the difference between two neighbouring cells must therefore mostly be a slight increase in relative humidity - a vertical




shift in Figure 1. If we lower $G$, both neighbouring cells could then be to the right of the limiting slope, preventing a persistent contrail from forming in the cell that is supersaturated with respect to ice and thus also preventing the border. In this case, our
analysis does not recognise formation being limiting because the formation boundary was not crossed. If we now consider a horizontally contiguous ice supersaturated region and an arbitrary aircraft then this implies that this aircraft will or will not form a contrail, independently of where it crosses the region boundary. Furthermore, at high values of $G$, the limiting slope acts much like a vertical boundary. Indeed, the freezing and H2C-04 formation limiting factors show very similar results (cf. Figure 3 and Supplementary Figure 4).

How then to view the competition between the persistence and droplet formation limiting factors? The persistence and droplet freezing limiting factors together create an upper bound to the potential persistent contrail formation region size. The droplet formation limiting factor then controls what proportion of this region is available for persistent contrails to form. Figure 6 shows that the level of control varies with latitude and altitude: at high altitudes, aircraft design has barely any influence.

### 435 3.6 Uncertainties and limitations

As with all climatological analyses and models, there are a number of limitations in the usage of our results. For example, consider the latitude-dependent results in Figure 6. In theory, more information could be gained by including two distinct extratropical and tropical responses. No persistent contrails are produced in the tropics at 350 hPa and essentially none are produced at 300 hPa, but if the globally averaged fits were to be used, persistent contrails would be assumed to form in these
regions. However, the results in the individual latitude bands also have a higher seasonal dependence. There is thus a trade-off in the level of detail that can be included if persistent contrail formation is to be climatologically modelled. Clearly, it would be inappropriate to analyse the climate impact of an aviation scenario that only considers aircraft flying in one latitude region using the globally fitted responses. Similarly, it would also be inappropriate to use the fits to analyse aircraft flying only in a single season. Instead, our results are useful for global aviation or aircraft-specific scenarios with a time step of at least one
year.

It is also important to underline the difference between persistent contrail formation and the resulting contrail cirrus coverage and climate impact. The results presented in this study are valid only for static persistent contrail formation, i.e. do not consider contrail spreading or temporal changes in the local ambient conditions: We assume a contrail to be persistent if the ambient conditions are supersaturated with respect to ice at the time of formation, however since the water vapour field is highly
heterogeneous and can vary rapidly with time, the contrail may not actually persist. We are not able to consider dynamic conditions such as the subsidence of the local air mass that may lead to contrail sublimation. It is also possible that persistent contrails formed in one grid cell spread to neighbouring cells due to mixing. Particularly interesting are neighbouring cells that are limited by the droplet formation limiting factor, i.e. are themselves not conducive for persistent contrail formation, but are supersaturated with respect to ice and are below the homogeneous freezing temperature. In a previous study, these regions
were defined as "reservoirs" for persistent contrails (Wolf et al., 2023a). Due to the vertical or horizontal spreading of contrails with time, it is thus possible that persistent contrails can nevertheless form in grid cells that our analysis found to be limited



by droplet formation. Such spreading is certainly important for the calculation of a contrail cirrus coverage, but since it is a secondary, dynamic effect we have not included it in this study on persistent contrail formation.

Care should also be taken when using our results to inform aircraft design or flight trajectories. For example, we show that replacing kerosene with hydrogen along assumed future flight trajectories could result in 71.4 % more persistent contrails (cf. Table 2) on a climatological timescale without considering contrail avoidance strategies. This does not, however, translate to a 71.4 % larger radiative forcing or temperature change due to differing contrail properties. For example, Rap et al. (2023) find a 70 % increase in contrail coverage from hydrogen-powered aircraft compared to conventional aircraft, largely in line with our results, but also find a 25 % reduction in contrail cirrus Effective Radiative Forcing (ERF). Similar results were found by Grewe

et al. (2017a). Persistent contrail formation should thus not be used as a direct proxy for climate impact, and more research is required into contrail properties from novel aviation fuels and propulsion technologies. We also reiterate that contrails are only part of the equation and that the full climate impact, including other non-$CO_2$ effects, should be used to inform aircraft design or flight trajectories.

    The globally averaged, climatological results presented in this study are not expected to change significantly due to a chang-

ing climate. Our results show that persistence is the most limiting factor for all aircraft designs. Therefore, the probability of persistent contrail formation should vary in tandem with the frequency and location of ISSRs. Climate models predict a generally warmer upper troposphere and generally cooler lower stratosphere, leading to a decrease in relative humidity in the tropics and an increase towards to poles (Benetatos et al., 2024; Irvine and Shine, 2015). This is reflected in the results of both Bock and Burkhardt (2019) and Chen and Gettelman (2016), who find for a future atmosphere a decrease in the probability

of persistent contrail formation (here $p_{pcf}$) in the tropics and an increase in the northern extratropics. Chen and Gettelman (2016) find a net reduction in contrail cirrus radiative forcing, whereas Bock and Burkhardt (2019) expect the radiative forcing to approximately even out.

    If the global $p_{pcf}$ remains approximately the same, then the limiting factor results (Figure 4) and the competition between the limiting factors should also remain unchanged. However, the regional differences in ISSR location may modify the shape of the responses in Figure 6, lowering the maximum $p_{pcf}$ values in the tropics and raising them in the extratropics. For the

pressure levels 250 to 200 hPa, which are driven by both a tropical and extratropical response, the global result will likely have a steeper initial slope and a shallower secondary slope, shifted slightly to higher mixing line slopes. Nevertheless, the maximum values and the general shape of the responses are not expected to change significantly.

    Finally, we do not expect the choice of global aviation scenario to have significantly affected the weighted results. In this

study, we used the DEPA 2050 progressive scenario for the year 2050 since it is representative of a future multi-fuel global fleet and due to its ease of publication. Other global scenarios may produce slightly different weighted results, but the focus should nevertheless remain on the northern extratropics at 300-250 hPa.





## 4  Conclusions

Persistent contrail formation is limited by three main factors: droplet formation, droplet freezing and persistence. Our inves-
tigation using ERA5 data in the 2010 decade finds persistence (ice supersaturation) to be the most limiting factor. Of the
considered grid cells, 92.3 % are not supersaturated with respect to ice and thus not conducive to persistent contrail formation.
Droplet freezing is found to be limiting 18.2 % of the time and droplet formation between 18.1 % (hydrogen-powered aircraft)
and 89.5 % (Water Enhanced Turbofan with 75 % water vapour reduction). The boundaries of persistent contrail formation
regions are also generally defined by the persistence limiting factor, regardless of aircraft design. In other words, should an
aircraft - without changing its operating condition - start producing a persistent contrail, generally it is because the air mass
it has entered is supersaturated with respect to ice. Our results thus underscore the importance of accurately estimating ice
supersaturated regions (Gierens et al., 2020; Hofer et al., 2024b).

We demonstrate the feasibility of developing high-quality climatological relationships between the mixing line slope $G$
and the potential persistent contrail formation $p_{pcf}$ for individual pressure levels. We create these relationships using the sum
of two modified logistic functions, one representing the response from the tropics and the other from the extratropics. The
combination of the persistence and droplet freezing limiting factors create an upper bound for the size of persistent contrail
formation regions, which varies by latitude, altitude and season. We find that using the standard Schmidt-Appleman Criterion,
aircraft design can no longer limit persistent contrail formation for mixing line slopes $G > 4.29$ Pa/K. However, due to the
distribution of ice supersaturation with altitude, the actual limit is generally lower and decreases with altitude. Therefore, the
higher the altitude, the less influence the aircraft design has. For very low $G$, persistent contrails form only at very high altitude.
As $G$ increases, more persistent contrails are also formed at lower altitudes. These climatological relationships are the first step
in the development of a computationally inexpensive method to analyse the contrail climate impact of novel aviation fuels and
propulsion technologies on a climatological timescale.

Novel aviation fuels and propulsion technologies will have a major impact on persistent contrail formation. Across all
considered flight levels, improving overall propulsion efficiency from 0.3 to 0.4 increases persistent contrail formation by
13.8 % along representative flight trajectories. Replacing all aircraft by hydrogen combustion or fuel cell aircraft could result
in a 71.4 % increase. We note, however, that persistent contrail formation, as analysed here, is not synonymous with contrail
climate impact. Contrails produced by aircraft using alternative fuels such as hydrogen are expected to have very different
optical and radiative properties. Although such aircraft may produce significantly more persistent contrails, their overall contrail
climate impact may in fact be lower than conventional aircraft (Bier et al., 2024; Rap et al., 2023; Gierens, 2021). More
research is required to better understand this effect. On the other hand, the Water Enhanced Turbofan concept could reduce
persistent contrail formation by 53.6 % to 85.6 % along expected flight trajectories (WET-50/WET-75 vs. CON-NG). Our
results, therefore, point to a clear potential for the targeted introduction of aircraft with low mixing line slopes to significantly
reduce persistent contrail formation.



*Code and data availability.* The Python code and the processed and compiled data used to perform all analyses are provided in the 4TU.ResearchData repository at https://doi.org/10.4121/cdb4e3bb-d6f4-4422-a715-b6187098a314. The DEPA air traffic scenarios for 2020 to 2050 are available from Zenodo at https://doi.org/10.5281/zenodo.11442323; ERA5 data from ECMWF at https://cds.climate.copernicus.eu/.

*Author contributions.* Conceptualisation: L.M. and V.G.; Methodology, investigation and visualisation: L.M.; Supervision: V.G.; Original draft: L.M.; Review and editing: L.M. and V.G. Both authors read and approved the final manuscript.

*Competing interests.* The authors declare no competing interests.

*Acknowledgements.* The authors thank Dr. Klaus Gierens for his suggestions and advice, Sina Hofer for providing data comparing ERA5 and MOZAIC/IAGOS and Dr. Luca Bugliaro for providing an internal review. The authors are thankful for the support of the German Climate Computing Center (DKRZ) under project Ökoluft (bd1062). The authors are supported by the 328H2-FC and DINA2030+ projects within the sixth Aviation Research Programme of the German Federal Government (LuFo VI-2). The funding bodies played no role in the study
design, data collection, analysis and interpretation of data, or the writing of the manuscript.



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
