# Peer review of "Investigating the limiting aircraft design-dependent and environmental factors of persistent contrail formation"

_EGUsphere, 2024_

## Referee Comment (RC1)

**Comments on "Investigating the limiting aircraft design-dependent and environmental factors of persistent contrail formation" by Liam Megill and Volker Grewe (https://doi.org/10.5194/egusphere-2024-3398)**

The manuscript investigates the influence of potential future aircraft-engine designs on the formation of condensation trails (contrails). The authors identify three main constraints on contrail formation: persistence (supersaturation), droplet formation, and droplet freezing. These factors are then set into context to aircraft-dependent designs determined by the type of propulsion system and fuel used. As the authors point out, alternative fuels such as hydrogen can cause more persistent contrails, while other technologies such as water enhanced turbofan engines can reduce persistent contrail formation compared to conventional kerosene jet engines.

The three limiting factors for persistent contrail formation and their combinations are identified on a global scale, but also weighted with future scenarios of air traffic in 2050, thus projecting into the future and showcasing contrail formation using future jet engines.
The statistical analysis is based on 10 years of three-dimensional ERA5 data, primarily using temperature and relative humidity fields. The ERA5 data are used to determine the location and frequency of occurrence of ice supersaturated regions (ISSR) and how often one or more of the above limiting factors restrict contrail formation. Limitations of the ERA5 data itself – underestimation of ISSR – are discussed and partially corrected for by the use of scaling factors. The result section further elaborates about the "spatial variability of persistent contrail formation" and "climatological potential persistent contrail formation".

The authors cite latest and relevant literature in the motivation / introduction. The presented results are also compared and set into context with current literature.
The manuscript is generally very well written and structured, with clear and readable figures. The manuscript fits into the scope of ACP and I recommend this manuscript for publication after the authors have satisfactorily addressed the major and minor comments listed below.

**Major comments:**
(1) The authors investigate and discuss the formation potential of persistent contrails in time and space using alternative engine types and fuels instead of today's kerosene jet engines. As shown in this manuscript and in previous studies, the use of hydrogen is expected to increase the number of persistent contrails. However, contrails exert a radiative forcing on short time scales of minutes to days, if they form at all. In contrast, fossil fuels and the release of carbon dioxide causes a radiative forcing on time scales of several decades and longer. The summary gives the impression that hydrogen powered aircraft have a generally negative impact on the climate. While it may be beyond the scope of the manuscript to calculate the actual radiative forcing, the authors should mention in the summary that "green" hydrogen, which is not sourced from fossil fuels, would cause more persistent contrails, but without the negative effect of CO2 emissions, and is therefore likely to be more environmentally friendly compared to conventional jet engines.

(2) One comment concerns the use of the supplementary material. In the manuscript the authors often refer to figures in the supplement. The authors might consider including the frequently and therefore seemingly relevant figures in the appendix or in the main text. Often

referenced figures are Figures 4, 5, and 6 from the supplement. These figures could be exchanged with Table 2, which provides detailed numbers but is not necessarily needed directly in the main text and could be shifted in the appendix or the supplement. Table 2 is partly redundant as the important numbers are also given and discussed in the main text.

(3) Regardless of comment 2 and where the figures are eventually located, ambiguity in the referenced figures could be avoided. For example, the authors refer to Figure 6 in the text and to Figure 6 in the supplement in one sentence. It would be clearer if the authors use: Figure 6 for figures in the main text, Figure A6 for figures in the appendix, and S6 for figures in the supplement.

**Minor comments:**

L16: Please explain CO2 at first appearance.

L21: Why use of capital letters for "Effective Radiative Forcing"? Is the abbreviation / acronym missing?

L60: Please explain "H/C ratio".

L69: Please add a space between the unit and "(kerosene)" as well as "(hydrogen)".

L103: Could you introduce the abbreviation "CON" earlier, at the first occurrence?

L106: Please check for consistent use of the serial/oxford comma. It was not used before and after, but here in this line.

L113: Abbreviations and symbols should be avoided at the beginning of a sentence.

L122: Do you mean "...we define \mean{$c\_p$} to be.."?

L123 and 124: Please check for grammar. Abbreviations and symbols at the beginning of a sentence should be avoided. This also applies to later occurrences in the text.

L138: Why is homogeneous freezing required? If ice nucleating particles are present, contrails could also form at temperatures above 235K? For example, in the triangle formed by the 235 K line, the G-line, and the water vapor saturation curve? The Schmidt-Appleman criterion requires homogeneous freezing, but in reality this would mean that contrails can form at higher ambient temperatures?

L144: Please explain "nvPM".

Table 1: Please explain (E) and (K) in row 3, column 7.

L161-162: Please explain 0.5 and 1.0 **V**, even though it can be guessed from the next sentence.

L168: "altitudes higher than 217 hPa" could be confusing. Suggest to re-write, e.g.: At pressure levels lesser than 217 hPa. This also applies to later occurrences in the text.

L187: Could you specify where you selected 2160 hours from. It implies that DEPA 2050 is an inventory of simulated flight tracks. The DEPA 2050 data could also be some kind of a weighting function. You might briefly mention the type of info of DEPA2050 data in section 2.3.

L208: You are interested in regions with supersaturation where RHi > 1, correct? Why are you focusing on RHi<1?

L305 to 307: The sentence "Since the horizontal…". What is meant with the word "aligned"? Spatially or statistically? A short explanation would be helpful.

Figure 4 y-axis label: "limfact". Even though the meaning could be guessed it would be good to write the full word in the label or to use an abbreviation that is explained in the figure caption.

L351: Please explain what is meant by "...*conservatively* regrid…"

L368: "north" instead of "above"?

L369: "south" instead of "below"?

L375: "aim" instead of "wish"?

L376: *G* already introduced

---

## Referee Comment (RC2)

**Comment on the "Investigating the limiting aircraft design-dependent and environmental factors of persistent contrail formation" by Liam Megill and Volker Grewe**

**General Comments:**

The study by Megill & Grewe examines the limiting factors related to aircraft design and environmental conditions in the formation of persistent condensation trails. The aircraft design factors considered include propulsion efficiency in relation to the type of fuel used (Sustainable Aviation Fuels - SAFs, hydrogen - $H_2$ fuel cells and combustion, hybrid electric aircraft, and the Water-Enhanced Turbofan - WET). The environmental conditions studied are ice supersaturation, droplet formation, and droplet freezing. The dataset used to calculate the environmental conditions is the ERA5 data set. The analysis is performed in a 3D dimensional framework using data from the decade of 2010. To limit computation cost and autocorrelation, the authors randomly selected data from a certain number of hours per season. The authors developed climatological relationships describing potential persistent contrail formation as a function of pressure level and Schmidt-Appleman mixing line slope G, and found that the influence of aircraft design on persistent contrail formation decreases with increasing altitude. They found that ice supersaturation is the most limiting factor for all aircraft designs considered. They also found that, on the one hand, globally averaged persistent contrail formation could increase by 13.8% for next-generation conventional aircraft, or by 71.4% if all aircraft were replaced with hydrogen combustion or fuel cell equivalents. On the other hand, water vapor extraction technologies, such as the Water Enhanced Turbofan concept, have the potential to reduce persistent contrail formation by 53.6% to 85.6%.

This is an excellent study, carefully produced and well-written. It fits perfectly within the CPA framework. I therefore recommend publication of this article once the authors have responded to the major and minor comments listed below.

**Majors comments:**

1. The authors investigate the limiting factors related to aircraft design and environmental conditions in the formation of persistent contrails using ERA5 data. Consistent with a previous study that reported biases in ERA5 relative humidity, the authors apply a bias correction approach to address this deficiency. However, previous studies have also reported biases in ERA5 temperature data (e.g. Wolf et al. 2023), in the upper troposphere and lower stratosphere. Since temperature is an essential parameter in the Schmidt-Appleman criterion, I recommend that the authors discuss the

potential impact of these biases on their results, since they didn't correct them as in Wolf et al. (2023).

2. The authors conducted their study over the decade of 2010. To limit computational costs and reduce autocorrelation, they randomly selected 10 % (one-year duration data) of the total data. It is questionable whether the study can be considered a climatological framework study, especially at the regional scale, where the limited number of sampling points may raise concerns about statistical robustness. The authors should perform a sensitivity test on the sample size (number of hours or percentage) to assess the robustness of the results (e.g., variation in the fraction of persistent contrails with fuel change) and the conclusions of the study.

**Minors:**

L18: Even if persistent contrails quickly dissipate, they can still be expected to have localized and short-term small impacts on the climate. It is preferable to use terms like "small" or "negligible impact", etc. instead of suggesting "no impact."

L18: Please start a new sentence from "In certain conditions"

L22: Please, cite the references in chronological order and ensure this is consistently applied throughout the manuscript.

L205: Please specify what $RHi_C$ is.

L89-90: Please split the sentences into two for easier understanding, starting the second sentence for example with "but in total prevents".

L95: Please define the acronym ERA5

L240 : Typos, please change the second "that" before the word "factor" to "the".

**Reference:**

Wolf et al. 2023. Correction of temperature and relative humidity biases in ERA5 by bivariate quantile mapping: Implications for contrail classification.

---

## Author Response (AR1)

**EGUSPHERE-2024-3398**

Peer Review - Megill and Grewe, "Investigating the limiting aircraft design-dependent and environmental factors of persistent contrail formation".

**Author Responses: First Round**

We thank the reviewers for their very valuable feedback. With their help, we believe that we were able to significantly improve the quality and impact of our work. We have addressed all comments on a point-by-point basis below and updated our manuscript accordingly. We have submitted a revised manuscript as well as a track-changes version, in which additions are indicated in blue and deletions in red. Unless specifically stated, our references to line numbers refer to the revised manuscript (without track-changes).

We would like to address two significant changes that we made to our manuscript.

1. **Updated methodology of the droplet formation limiting factor**
   Since uploading the pre-print, we have reconsidered the issue briefly discussed in ln. 144-148 of the original pre-print: With our original methodology, we were unable to capture the lower "effective freezing temperature", notably shown by Bier et al. (2022, 2024) for both conventional as well as hydrogen combustion aircraft. This resulted in an overestimation of the potential persistent contrail formation for high-$G$ aircraft and an asymptote at $G > 4.29$ Pa/K.

   What we did not consider in the original methodology, is that for ice crystals to form, the plume must be *concurrently* supersaturated with respect to water and below the homogeneous freezing temperature. If the plume is subsaturated with respect to water again before the homogeneous freezing temperature is reached, the water droplets would likely evaporate before they could freeze (see e.g. Figure 3 of Bier et al., 2024). Consider Figure 1 of the original pre-print: For an aircraft with $G = 3.7$ Pa/K at ambient conditions defined by the green point for droplet formation, the mixing line would drop below the water saturation curve before it crosses the 235.15 K boundary. We thus realised that defining $T_{max}$ as the temperature at which the threshold mixing line is tangential to the water saturation curve is insufficient for high-$G$ aircraft.

   Instead, we suggest modifying the definition of $T_{max}$ such that it is equal to 235.15 K when it would otherwise be greater than 235.15 K (see Eq. (5) of the new manuscript). This occurs for $G > 2.38$ Pa/K. In this way, we ensure that all possible mixing lines are at some point concurrently supersaturated with respect to water and colder than 235.15 K (see the new Figure 1 for $G = 6.0$ Pa/K). Using this method, we were also able to closely replicate the ambient temperatures at which Bier et al. (2022 & 2024) found ice crystal numbers of zero (their Figures 5(c) and 4 respectively). We show this in the new Figure S1.

   We appreciate that this - much like the Schmidt-Appleman Criterion in general - is only an approximation and that, in reality, the formation of contrails depends on many microphysical processes. The mixing process does not, in reality, follow the straight lines shown in Figures 1 and S1. However, we believe that this change already goes a long way in improving the standard definition of the SAC for high-$G$ aircraft.

   *Influence on the results:* The overall results are not very much affected by this change. The largest difference is in the fitting of the potential persistent contrail formation as a function of the mixing line slope $G$ (Figure 7 and Table 2) at high values of $G$ (hydrogen-powered aircraft).

2. **Overall propulsion system efficiency of reference aircraft**
   We received a comment from a colleague that our initial choice of overall propulsion system efficiency for the reference aircraft CON-LG ($\eta = 0.30$) is low compared to the current state-of-the-art. This slightly affected the reported percentage increases/decreases of different technologies compared to conventional aircraft. We have, therefore, decided to use a state-of-the-art overall propulsion system efficiency of 0.37 as our reference aircraft (CON-37). We have also tried to always state the reference aircraft when reporting percentage increases/decreases.

*Influence on the results:* The overall results are not affected by this change, only how they are presented. As a result of increasing the overall propulsion system efficiency, the increase in $p_{pcf}$ due to more efficient engines is slightly reduced. The reduction in the globally-averaged $p_{pcf}$ increase from hydrogen-powered aircraft is mostly due to the change made to the definition of $T_{max}$ (see above), but is also influenced by the change in reference aircraft.

We hope that we have properly understood all comments and feedback and welcome any further comments, suggestions or points of clarification.
* * *
**Responses to Reviewer #1**

✏️ **Comment R1.1**

The manuscript investigates the influence of potential future aircraft-engine designs on the formation of condensation trails (contrails). The authors identify three main constraints on contrail formation: persistence (supersaturation), droplet formation, and droplet freezing. These factors are then set into context to aircraft-dependent designs determined by the type of propulsion system and fuel used. As the authors point out, alternative fuels such as hydrogen can cause more persistent contrails, while other technologies such as water enhanced turbofan engines can reduce persistent contrail formation compared to conventional kerosene jet engines.

The three limiting factors for persistent contrail formation and their combinations are identified on a global scale, but also weighted with future scenarios of air traffic in 2050, thus projecting into the future and showcasing contrail formation using future jet engines. The statistical analysis is based on 10 years of three-dimensional ERA5 data, primarily using temperature and relative humidity fields. The ERA5 data are used to determine the location and frequency of occurrence of ice supersaturated regions (ISSR) and how often one or more of the above limiting factors restrict contrail formation. Limitations of the ERA5 data itself – underestimation of ISSR – are discussed and partially corrected for by the use of scaling factors. The result section further elaborates about the "spatial variability of persistent contrail formation" and "climatological potential persistent contrail formation".

The authors cite latest and relevant literature in the motivation / introduction. The presented results are also compared and set into context with current literature. The manuscript is generally very well written and structured, with clear and readable figures. The manuscript fits into the scope of ACP and I recommend this manuscript for publication after the authors have satisfactorily addressed the major and minor comments listed below.

**Response to R1.1:** Thank you for taking the time to help us improve our manuscript. Your comments are greatly appreciated and led us to some very interesting and crucial further findings.
* * *
✏️ **Major Comment R1.2**

The authors investigate and discuss the formation potential of persistent contrails in time and space using alternative engine types and fuels instead of today's kerosene jet engines. As shown in this manuscript and in previous studies, the use of hydrogen is expected to increase the number of persistent contrails. However, contrails exert a radiative forcing on short time scales of minutes to days, if they form at all. In contrast, fossil fuels and the release of carbon dioxide causes a radiative forcing on time scales of several decades and longer. The summary gives the impression that hydrogen powered aircraft have a generally negative impact on the climate. While it may be beyond the scope of the manuscript to calculate the actual radiative forcing, the authors should mention in the summary that "green" hydrogen, which is not sourced from fossil fuels, would

cause more persistent contrails, but without the negative effect of CO2 emissions, and is therefore likely to be more environmentally friendly compared to conventional jet engines.

**Response to R1.2:** Thank you for this comment. We agree that the impression that hydrogen-powered aircraft would have a generally negative impact on the climate should be avoided. This is of course complex and the topic of ongoing research. Therefore, we have modified the end of the summary to highlight that further work is required to translate changes in persistent contrail formation to changes in climate impact. We have also in many locations, including the title and summary, clearly stated that this study only concerns persistent contrail formation.

However, we do not believe that there needs to be a reference to green or any other coloured hydrogen, nor to CO2 emissions. We believe that this would distract from this article's story, since the production method of the hydrogen does not influence persistent contrail formation.
* * *
**✎ Major Comment R1.3**

One comment concerns the use of the supplementary material. In the manuscript the authors often refer to figures in the supplement. The authors might consider including the frequently and therefore seemingly relevant figures in the appendix or in the main text. Often referenced figures are Figures 4, 5, and 6 from the supplement. These figures could be exchanged with Table 2, which provides detailed numbers but is not necessarily needed directly in the main text and could be shifted in the appendix or the supplement. Table 2 is partly redundant as the important numbers are also given and discussed in the main text.

**Response to R1.3:** Thank you for this suggestion. We agree that the placement of figures was not optimal. We have moved the previous Figure S4 (now Figure 4) to the main text and moved the previous Table 2 to the supplement, where it has replaced the previous Table S1. However, we do not believe that Figures S5 and S6 are important enough for the story to move them to the main text.
* * *
**✎ Major Comment R1.4**

Regardless of comment 2 and where the figures are eventually located, ambiguity in the referenced figures could be avoided. For example, the authors refer to Figure 6 in the text and to Figure 6 in the supplement in one sentence. It would be clearer if the authors use: Figure 6 for figures in the main text, Figure A6 for figures in the appendix, and S6 for figures in the supplement.

**Response to R1.4:** We agree that this is clearer. We have updated the references as suggested.
* * *
**✎ Comment R1.5**

L16: Please explain CO2 at first appearance.

**Response to R1.5:** We have defined the first instance of CO2 (ln. 17).
* * *
**✎ Comment R1.6**

[Figure]

L21: Why use of capital letters for "Effective Radiative Forcing"? Is the abbreviation / acronym missing?

**Response to R1.6:** Thank you for pointing this out. We have removed the capitalisation.

[Figure]
 **Comment R1.7**

L60: Please explain "H/C ratio".

**Response to 1.7:** We have replaced "H/C ratio" with its descriptor: "hydrogen-to-carbon ratio". This is indeed more clear.

**Comment R1.8**

L69: Please add a space between the unit and "(kerosene)" as well as "(hydrogen)".

**Response to R1.8:** Changed as suggested.

**Comment R1.9**

L103: Could you introduce the abbreviation "CON" earlier, at the first occurrence?

**Response to R1.9:** We unfortunately do not fully understand what is meant by this comment. We introduce the abbreviation "CON" directly before it is used in Eq. (1) and do not use it before this stage. If this comment refers to the first mention of "conventional kerosene", then we would argue that we would have to re-define "CON" immediately before Eq. (1) again anyway since, in comparison to SAF or H2, it is not a common abbreviation. Therefore, we believe introducing the abbreviation "CON" at an earlier stage would not help the reader understand the study. The more important definition of the abbreviation occurs in Section 2.2, where the aircraft IDs are described.

**Comment R1.10**

L106: Please check for consistent use of the serial/oxford comma. It was not used before and after, but here in this line.

**Response to R1.10:** We have checked the manuscript and removed the remaining Oxford commas for consistency.

**Comment R1.11**

L113: Abbreviations and symbols should be avoided at the beginning of a sentence.

**Response to R1.11:** We assume that this comment refers to the sentence defining $Q_E^0$. Thank you for pointing this out. We have now included its definition, taken from Yin et al. (2020), to be the "quasi-electric energy content". We have

also checked the rest of the manuscript to ensure that no sentences begin with a symbol.
* * *
**✏ Comment R1.12**

L122: Do you mean "...we define \mean{c_p} to be.."?

**Response to R1.12:** This section was indeed not clear, thanks for bringing this to our attention. We have re-formulated it to: "We make use of the mol-based heat capacity of the exhaust gases $\bar{c}_p$, which Gierens (2021) showed not to be a constant (see their Eq. (15)). Nevertheless, we assume it to be $30.6\,\mathrm{J/mol/K}$ for all pressure levels for simplicity" (ln. 123–125).
* * *
**✏ Comment R1.13**

L123 and 124: Please check for grammar. Abbreviations and symbols at the beginning of a sentence should be avoided. This also applies to later occurrences in the text.

**Response to R1.13:** We have re-formulated this and later occurrences to avoid symbols at the beginning of sentences.
* * *
**✏ Comment R1.14**

L138: Why is homogeneous freezing required? If ice nucleating particles are present, contrails could also form at temperatures above 235K? For example, in the triangle formed by the 235 K line, the G-line, and the water vapor saturation curve? The Schmidt-Appleman criterion requires homogeneous freezing, but in reality this would mean that contrails can form at higher ambient temperatures?

**Response to R1.14:** We unfortunately do not fully understand this comment, but we believe it is in line with the change we made to the overall methodology. For a contrail to form, the mixture must be supersaturated with respect to water such that water droplets form. Then, these water droplets must freeze, which we can thermodynamically assume to occur at the homogeneous freezing temperature of 235.15 K. Both of these conditions must occur simultaneously. Ergo, a mixing line just to the left of the blue line shown in our original Figure 1 would not result in contrail formation, since the mixture would be subsaturated with respect to water by the time it was below 235.15 K. We do not assume that contrails can form at ambient temperatures above 235.15 K.
* * *
**✏ Comment R1.15**

L144: Please explain "nvPM".

**Response to R1.15:** This definition was overlooked, thank you for pointing it out. We have added its definition as suggested.
* * *
**✏ Comment R1.16**

> Table 1: Please explain (E) and (K) in row 3, column 7.

**Response to R1.16:** We do not believe that this is necessary since all values included in this table are inputs to the equations (1) – (4), where each variable is described. We have instead mentioned in the Table caption that all variables are described in the text. We hope that this is sufficient.
* * *
> ✎ **Comment R1.17**
>
> L161–162: Please explain 0.5 and 1.0 V, even though it can be guessed from the next sentence.

**Response to R1.17:** Amended as suggested.
* * *
> ✎ **Comment R1.18**
>
> L168: "altitudes higher than 217 hPa" could be confusing. Suggest to re-write, e.g.: At pressure levels lesser than 217 hPa. This also applies to later occurrences in the text.

**Response to R1.18:** Thank you for this comment. We agree that this may be deemed confusing. This particular paragraph has since been removed, but we have checked the remainder of the manuscript for similar wording. We have introduced the Flight Level (FL) when mentioning the altitude specifically, but always include the corresponding pressure level for consistency. Where necessary, we have thus included the change in altitude alongside the change in pressure level, e.g. "at altitudes below FL300 (pressure levels $\geq$ 300 hPa)" (ln. 312–313). We believe that this is a better solution that should be free from misinterpretation.
* * *
> ✎ **Comment R1.19**
>
> L187: Could you specify where you selected 2160 hours from. It implies that DEPA 2050 is an inventory of simulated flight tracks. The DEPA 2050 data could also be some kind of a weighting function. You might briefly mention the type of info of DEPA2050 data in section 2.3.

**Response to R1.19:** Thank you for this comment. The value 2160 is actually a typo, also present in our code. It should be 2190, which is approximately 10% of the total hours within one season. There are $24 \times 365 \times 10 = 87600$ hours within the 2010 decade (ignoring leap years for simplicity). Assuming each season has the same number of hours, which is approximately correct except for DJF, the total number of hours per season would be 21900, which then becomes 2190 when only considering 10% of the data. We have since corrected the typo and performed the missing simulations.

Additionally, we have added more detail about the DEPA 2050 emission inventory to section 2.3: "We use the forecasted aggregate yearly fuel and distance flown by the global fleet of aircraft in the year 2050 from the DEPA 2050 progressive scenario, which is available with a 1° latitude resolution. The emission inventory is included in the linked data and is shown graphically in Figure S2." (ln. 186–189).
* * *
> ✎ **Comment R1.20**

> L208: You are interested in regions with supersaturation where RHi > 1, correct? Why are you focusing on RHi < 1?

**Response to R1.20:** Thank you for this comment. In this study, we are not interested in the magnitude of the supersaturation, only whether or not the ambient conditions *are* supersaturated. We focus on RHi < 1 here because, as we note in the paper, ERA5 generally underestimates the relative humidity (see also the cited studies and Hofer et al., 2024). Therefore, we use the correcting factors, as in previous studies, to increase the ERA5 relative humidity values and give a distribution that more accurately reflects that of e.g. IAGOS. Supplementary Figure 3 shows that $RHi_{cor} = 0.95$ comes closest to resembling the MOZIAC/IAGOS CDF found by Hofer et al. (2024) at RHi = 1.0. This is meaningful because it then suggests that with this correction, we come close to having the same number of sub/supersaturated conditions in the dataset as in MOZIAC/IAGOS. How the CDF looks for RHi > 1 is not relevant for this study, since we are only concerned about persistent contrail formation and not, for example, contrail-cirrus coverage.
* * *
**✎ Comment R1.21**

> L305 to 307: The sentence "Since the horizontal…". What is meant with the word "aligned"? Spatially or statistically? A short explanation would be helpful.

**Response to R1.21:** We have amended the sentence to make clear that we have only adjusted the y-axis limits to visually align the CON-37 total limiting factor (ln. 325–327). This is done such that the general shapes of the two analyses (horizontal and vertical) can be compared to one another. A direct comparison is not possible because of the different units.
* * *
**✎ Comment R1.22**

> Figure 4 y-axis label: "limfact". Even though the meaning could be guessed it would be good to write the full word in the label or to use an abbreviation that is explained in the figure caption.

**Response to R1.22:** Thanks for noticing this. We have updated the figure labels with the full description, rather than the abbreviation.
* * *
**✎ Comment R1.23**

> L351: Please explain what is meant by "…conservatively regrid…"

**Response to R1.23:** With "conservative regridding" we are referring to conserving the same total sum from the source to the destination grid. We have added the note "(conserving the total sum of distance flown)" to the text (ln. 371).
* * *
**✎ Comment R1.24**

> L368: "north" instead of "above"?

**Response to R1.24:** Amended as suggested.
* * *
**✎ Comment R1.25**

L369: "south" instead of "below"?

**Response to R1.25:** Amended as suggested.
* * *
**✎ Comment R1.26**

L375: "aim" instead of "wish"?

**Response to R1.26:** Amended as suggested.
* * *
**✎ Comment R1.27**

L376: G already introduced

**Response to R1.27:** True, thanks for pointing this out. We have removed the extra description of $G$.
* * *
**Responses to Reviewer #2**

**✎ Comment R2.1**

The study by Megill & Grewe examines the limiting factors related to aircraft design and environmental conditions in the formation of persistent condensation trails. The aircraft design factors considered include propulsion efficiency in relation to the type of fuel used (Sustainable Aviation Fuels - SAFs, hydrogen - $H_2$ fuel cells and combustion, hybrid electric aircraft, and the Water-Enhanced Turbofan - WET). The environmental conditions studied are ice supersaturation, droplet formation, and droplet freezing. The dataset used to calculate the environmental conditions is the ERA5 data set. The analysis is performed in a 3D dimensional framework using data from the decade of 2010. To limit computation cost and autocorrelation, the authors randomly selected data from a certain number of hours per season. The authors developed climatological relationships describing potential persistent contrail formation as a function of pressure level and Schmidt-Appleman mixing line slope G, and found that the influence of aircraft design on persistent contrail formation decreases with increasing altitude. They found that ice supersaturation is the most limiting factor for all aircraft designs considered. They also found that, on the one hand, globally averaged persistent contrail formation could increase by 13.8% for next-generation conventional aircraft, or by 71.4% if all aircraft were replaced with hydrogen combustion or fuel cell equivalents. On the other hand, water vapor extraction technologies, such as the Water Enhanced Turbofan concept, have the potential to reduce persistent contrail formation by 53.6% to 85.6%.

This is an excellent study, carefully produced and well-written. It fits perfectly within the CPA framework. I therefore recommend publication of this article once the authors have responded to the major and minor comments listed below.

**Response to R2.1:** Thank you for taking the time to help us improve our manuscript. Your comments are greatly appreciated.
* * *
✏️ **Major Comment R2.2**

The authors investigate the limiting factors related to aircraft design and environmental conditions in the formation of persistent contrails using ERA5 data. Consistent with a previous study that reported biases in ERA5 relative humidity, the authors apply a bias correction approach to address this deficiency. However, previous studies have also reported biases in ERA5 temperature data (e.g. Wolf et al. 2023), in the upper troposphere and lower stratosphere. Since temperature is an essential parameter in the Schmidt-Appleman criterion, I recommend that the authors discuss the potential impact of these biases on their results, since they didn't correct them as in Wolf et al. (2023).
* * *
**Response to R2.2:** Thank you for this comment. We are aware of two previous studies that investigated the difference in ambient temperatures between ERA5 and IAGOS: Wolf et al. (2025) – the now published version of Wolf et al. (2023) – and Hildebrandt (2024). From the probability density function in Wolf et al. (2025), for example their Figure 3, we can see that ERA5 both under- and over-estimates the temperature, depending on the pressure level. For the pressure levels 250 - 200 hPa, the average difference is between -0.1 and -0.7 K, meaning that ERA5 on average underestimates the temperature in comparison to IAGOS data. Their Appendix C further shows that there is little dependence of the temperature difference on latitude and longitude. As the altitude increases (pressure level reduces), ERA5 underestimates the temperature more.

We did not initially investigate any corrections to the temperature, because we assumed that this bias would not play a big role in the results. However, following this comment, we introduced a temperature correction and re-ran the climatological potential persistent contrail formation study. We used a global increase in ERA5 temperatures of 0.1, 0.5 and 1.0 K. We are not aware of previous studies that have employed a simple global increase in temperature and note that this is a major simplification. Nevertheless, it should provide an order of magnitude understanding of what effect the bias could have on the results.

We show the results in the figure below (now also in the supplement as Figure S7) – **(a)** uncorrected, **(b)** 0.1 K increase, **(c)** 0.5 K increase and **(d)** 1.0 K increase. We find that the effect of the temperature bias is significantly lower than that of relative humidity. There is very little discernible visual difference between the uncorrected and corrected results. Most notable is the slight reduction in $p_{pcf}$ at 250 hPa (pink).

[Figure]

Due to limited computational resources, we are unable to re-run the full limiting factors study with temperature corrections. If there is an effect, it will likely influence the droplet formation limiting factor at pressure levels higher than 225 hPa (altitudes below FL360). However, given the small change to the above response, we assume that the effect to the limiting factors study would also be minimal. We have included a paragraph on ambient temperature corrections in Section 2.4 (ln. 218–224).
* * *
✎ **Major Comment R2.3**

The authors conducted their study over the decade of 2010. To limit computational costs and reduce autocorrelation, they randomly selected 10 % (one-year duration data) of the total data. It is questionable whether the study can be considered a climatological framework study, especially at the regional scale, where the limited number of sampling points may raise concerns about statistical robustness. The authors should perform a sensitivity test on the sample size (number of hours or percentage) to assess the robustness of the results (e.g., variation in the fraction of persistent contrails with fuel change) and the conclusions of the study.

**Response to R2.3**: For the limiting factors study, we use 10% of all hours within the 2010 decade. On average, therefore, we consider full global data every 10 hours, which we believe is more than sufficient for a climatological study.

We have performed a further analysis to ensure that this is indeed the case. Since the climatological response functions (Figure 7) are significantly less computationally intensive, they are based on *all* hours within the 2010 decade. therefore, we can plot the potential persistent contrail formation for each aircraft in the limiting factors study, using 10% of the data, on top of the fitted responses in Figure 7 - see the figure below. Showing all seasonal values would be too cluttered for this analysis.

[Figure]

We can clearly see that the results using only 10% of the data (circular markers) closely resemble the full results (dashed lines). We also note that the dashed lines correspond to the fits (not the average) and thus a difference can be expected. We therefore believe that 10% of the data is sufficient for a climatological study.
* * *
✎ **Comment R2.4**

L18: Even if persistent contrails quickly dissipate, they can still be expected to have localized and short-term small impacts on the climate. It is preferable to use terms like "small" or "negligible impact", etc. instead of suggesting "no impact."

**Response to R2.4**: This is a good point, thanks for raising it. We have updated the wording to include "negligible impact" as suggested.
* * *
✎ **Comment R2.5**

L18: Please start a new sentence from "In certain conditions"

**Response to R2.5**: We have split the sentence as suggested into two: "Most contrails quickly dissipate and have a negligible impact on the climate. However, in certain ambient conditions, contrails can spread to form contrail-cirrus clouds and persist for many hours (Haywood et al., 2009; Schumann and Heymsfield, 2017)." (ln. 18-20).
* * *
✎ **Comment R2.6**

L22: Please, cite the references in chronological order and ensure this is consistently applied throughout the manuscript.

**Response to R2.6**: We have modified the order of the references as suggested.

✏ **Comment R2.7**

L205: Please specify what $RHi_C$ is.

**Response to R2.7:** We did indeed not specify what this factor is, thank you for pointing this out. We have renamed $RHi_C$ to $RHi_{cor}$ and described it as a "correcting factor" (ln. 212). We believe it should now be clear that this factor is used to correct the relative humidity with respect to ice.

✏ **Comment R2.8**

L289-90: Please split the sentences into two for easier understanding, starting the second sentence for example with "but in total prevents".

**Response to R2.8:** We have split the sentences into two as suggested.

✏ **Comment R2.9**

L95: Please define the acronym ERA5

**Response to R2.9:** Amended as suggested.

✏ **Comment R2.10**

L240 : Typos, please change the second "that" before the word "factor" to "the".

**Response to R2.10:** Thanks for this comment. The first "that" is a conjunction; the second "that" refers to the "given limiting factor", i.e. a specific limiting factor. We believe that this is clearer than using "the".

**Responses to Reviewer #3**

✏ **Comment R3.1**

This study uses 10 years of ERA5 data, along with existing thermodynamic theory on contrail formation and persistence, to investigate the probability of persistent contrail formation for different aircraft/engine designs and the factors that constrain it. Additionally, relationships between the mixing line slope (part of the thermodynamic theory of contrail formation) and persistent contrail formation probability are derived using the study's data, which are highly relevant to the assessment of novel technologies in the context of aviation's climate impact.

The paper is quite well-written, and the visualizations are mostly excellent. The methodology is sound. I feel like some of the findings (e.g. ice supersaturation is the most limiting factor, but less so for technologies with steeper mixing lines) could have been approximated through physical reasoning and already available climatologies on temperature and humidity. The study's results are still highly valuable, but I would have liked to see some hypotheses / comparisons based on less refined analyses to better appreciate the merits of the approach taken here.

In my opinion, the most valuable results of the paper are the "simple" relationships derived for the probability of potential persistent contrail formation as a function of the mixing line slope. I expect these to be incredibly useful within a tool such as AirClim, in order to assess the impact of changes to aircraft fleets.

Overall, the study is within the scope of the journal, presents novel findings, and is well-written. I therefore recommend its acceptance subject to addressing my comments below.

**Response to R3.1:** Thank you for taking the time to help us improve our manuscript. Your comments are greatly appreciated.
* * *
**✎ Major Comment R3.2**

Given that the results of this study are likely going to be used in climate impact assessments, I would appreciate more comparisons of the obtained results to observations or other studies. For example, how do the numbers in Figure 3 compare to observations from e.g. IAGOS? Does the occurrence of ice supersaturation found in ERA5 (potentially limited to an appropriate geographical region) match that found by IAGOS?

**Response to R3.2:** Thank you for this suggestion. The comparison of ERA5 with observations, for example from IAGOS, has already been sufficiently done by other groups (e.g. Reutter et al. (2020), Agarwal et al. (2022), Hofer et al. (2024) and Wolf et al. (2024, 2025)). A further analysis is unfortunately out of scope for this study.

We believe that we have sufficiently covered ERA5's biases compared to observations. As we discuss in the article, the most critical aspect is the estimation of ice supersaturated regions. There have been many suggestions on how to correct ERA5 to more accurately replicate in situ measurements such as IAGOS. We decided, in line with previous studies, to use a simple approach that applies a correcting factor of $1/RHi_{cor}$. As we show in Figure S4, $RHi_{cor} = 0.95$ provides a good fit against a global IAGOS CDF for RHi $\leq$ 1.0. Since the results of our work that could be used for climate impact assessments (e.g. the $p_{pcf}$ fits) also have a global resolution, we believe that this simple correction adequately covers the ERA5 RHi bias.
* * *
**✎ Major Comment R3.3**

There are several discussions in the paper regarding the way in which persistent contrail formation regions change due to modifying certain parameters. See for example the paragraph starting at L327. A figure (or two) illustrating such a situation would really help the reader, and also serve as a test for the hypotheses formed here and there.

**Response to R3.3:** Thank you for this comment. As we mention in the response to R3.4, we experimented with different visualisations and figures, some of which can be seen in the data repository. We believe that we have struck a good balance in the figures that we now show. We recognise that the figures are abstracted to various degrees (0D: global sum; or 2D: latitude & altitude). However, this is necessary because finding individual instances of region shapes etc. is not feasible due to the large variation in and temporal and spatial resolution of the data. Relating specifically to the discussion beginning at the old line 327, we believe that including a figure showing two (or more) regions coalescing and thus reducing their total perimeter would be superfluous since it is not a main outcome of the analysis. See also our response to R3.50.
* * *
**✎ Comment R3.4**

Did the authors experiment with visualizing persistent contrail formation regions as the intersection of 3 different regions (each satisfying one of the individual constraints)? Perhaps this could be a nice plot to add to figure 2.

**Response to R3.4:** Thank you for this suggestion! We did indeed experiment with various visualisations. We had some difficulty making a figure depicting each limiting factor individually that was not too cluttered to be useful. However, we were able to amend Figure 2 as suggested to show the individual limiting factors. We believe this aids the reader in understanding the analysis performed for the individual limiting factors.
* * *
[Figure]
 **Comment R3.5**

The abbreviation "limfac" is used in many figures, but I do not believe it is defined anywhere.

**Response to R3.5:** Thanks for spotting this. We have modified the figure labels accordingly.
* * *
[Figure]
 **Comment R3.6**

Many occasions where citations are not in chronological order: these should be re-ordered.

**Response to R3.6:** Modified as suggested.
* * *
[Figure]
 **Comment R3.7**

Many occasions where a "the" is missing before "horizontal direction" or "vertical direction". I've included some of those in my line-by-line comments below, but at some points I stopped marking them.

**Response to R3.7:** Added missing definitive articles as suggested.
* * *
[Figure]
 **Comment R3.8**

L18: Would replace "no effect" with "little effect"

**Response to R3.8:** Modified to "negligible impact".
* * *
[Figure]
 **Comment R3.9**

L20: should this be "resulting aircraft-induced cloudiness"?

**Response R3.9:** Agreed - amended as suggested.
* * *
**✎ Comment R3.10**

L21: effective radiative forcing (so first letters not capitalized). Same goes for the abstract.

**Response to R3.10:** Both have been corrected as suggested.
* * *
**✎ Comment R3.11**

L23: isn't "alternative fuels" descriptive enough?

**Response to R3.11:** We have removed the subclause such that it now reads: "As aircraft using alternative fuels are proposed and developed, [...]" (ln. 24).
* * *
**✎ Comment R3.12**

L25: "contrails **can** form"

**Response to R3.12:** Amended as suggested.
* * *
**✎ Comment R3.13**

L26: "ice nuclei" if the water vapour condensed on these particles, doesn't that make them cloud condensation nuclei and not necessarily ice nuclei?

**Response to R3.13:** Thank you for this comment. We have amended "ice nuclei" to "condensation nuclei" in line with previous studies (ln. 27–28).
* * *
**✎ Comment R3.14**

L36: "Contrails slowly sink". This is not necessarily true, I believe. There are processes that can lead to contrails moving upward, such as radiative heating.

**Response to R3.14:** Thank you for this comment. Whether or not contrails sink or rise is not relevant to the point we are trying to make, which is simply that persistent contrails spread out and form contrail cirrus. Therefore, in conjunction with the next comment, we have reformulated this sentence to: "These contrails slowly spread out, transitioning into contrail cirrus that can merge with natural cirrus and be transported over large distances (Kärcher, 2018)." (ln. 37–38).
* * *
**✎ Comment R3.15**

L36: "and often mixing with natural cirrus". Could you provide a reference for this statement?

Response to R3.15: Thank you for this comment. We have added Kärcher (2018) as a reference, who also references a number of other relevant studies (Immler et al., 2008). In conjunction with the previous comment, we have reformulated this sentence (see R3.14).
* * *
✏ **Comment R3.16**

L37: "trap outgoing longwave radiation". It is more physically correct to state "reduce outgoing longwave radiation", I believe.

Response to R3.16: Amended as suggested.
* * *
✏ **Comment R3.17**

L39: Not clear whether the warming effect dominates because of many night-time contrails or because day-time contrails are more warming than cooling, or a combination of both.

Response to R3.17: This is an interesting point. In global studies such as those by Burkhardt, Gettelman etc., everything is generally combined, so as far as we can tell it is not possible to comment on which effect is more relevant/dominant. The fact that contrails could, for example, be created during the night and convected into the sunlight, does not make things any simpler. On average, however, persistent contrails do cause a net warming. We therefore believe that the sentence as it currently stands is sufficient (ln. 40-41).
* * *
✏ **Comment R3.18**

L41: "over-proportional" should not be hyphenated I believe. I also think the wording could be better, perhaps by simply stating the results from the cited studies (e.g. x% of flights contribute y% of the total contrail radiative forcing…)

Response to R3.18: Thank you for this comment. We have rephrased this sentence to "Recent studies have shown that a small number of flights can have an outsize contribution to the total warming from contrail cirrus […]" (ln. 42-43). We do not think mentioning the values found by other studies is necessary since we are only mentioning big hits and contrail avoidance to put our study into context.
* * *
✏ **Comment R3.19**

L45: there are other studies that could be cited here, such as Agarwal et al. (2022) and Geraedts et al. (2024)

Response to R3.19: We do not believe these studies are relevant at this specific point in our manuscript. The Agarwal et al. (2022) study refers to reanalysis simulations such as ERA5 and MERRA-2, which is an important topic and very relevant to this work. The Geraedts et al. (2024) study develops an automated detection and matching system that can determine whether a persistent contrail forms. Neither of these studies relate directly to the difficulty in forecasting ice supersaturated regions, which is the topic of the sentence.

**Comment R3.20**

L47: should "the persistent contrail formation" be "persistent contrail formation" ?

**Response to R3.20:** We have amended the sentence to "[...] to the formation of persistent contrails, [...]" (ln. 48).

**Comment R3.21**

L48: what is "individual avoidance"?

**Response to R3.21:** We have specified this to "individual contrail avoidance" (ln. 49).

**Comment R3.22**

L51: "vertical" and "zonal" might be better terms to use here?

**Response to R3.22:** We already used "vertical" when referring to direction of the limiting factors analysis, and to us "zonal" would also imply a dependence on longitude, which we do not consider. We believe that "altitude" and "latitude" is much clearer.

**Comment R3.23**

L58: This section is somewhat at odds with an earlier section which states *"However, due to the large variability in the properties and concentration of the ambient aerosols (e.g. Brock et al., 2021; Voigt et al., 2022), the resulting ice crystal number and radiative effect of the subsequent contrails are **currently highly uncertain.**"*

**Response to R3.23:** We do not believe that this section is at odds with that earlier statement. This section really only adds more uncertainties. On the one hand, higher hydrogen-to-carbon ratios and lower aromatic content lead to lower soot and ice crystal number concentrations, and thus to a smaller optical depth and lower contrail climate impact. On the other hand, however, the higher hydrogen-to-carbon ratio leads to more frequent contrail formation (as our study also shows). We make no claim as to the net result regarding the radiative effect of contrails produced by SAF (or H2), because, as the previous section mentions, this is currently highly uncertain.

**Comment R3.24**

L82: "hybrid electric" this is hyphenated in most other places in the manuscript

**Response to R3.24:** Thanks for noticing this. We added the hyphen to match the rest of the manuscript.

**Comment R3.25**

L96: is this sentence missing a "are described"?

**Response to R3.25:** We removed the semi-colon and created a new sentence for better readability.

[Figure]
 **Comment R3.26**

L99: The introduction actually states that this is due to expansion of the exhaust plume, which is not necessarily the same as isobaric mixing I believe.

**Response to R3.26:** We have removed references to "expansion" and instead focused on "mixing".

[Figure]
 **Comment R3.27**

L100: How is this "thermodynamic approximation" if you describe the mixing process as isobaric?

**Response to R3.27:** By "thermodynamic approximation", we are referring to the Schmidt-Appleman theory, as envisaged by Schumann (1996). It is an approximation because contrail formation is governed by many microphysical processes (e.g. Kärcher et al., 2015). The theory assumes adiabatic and isobaric conditions (see e.g. Schumann, 1996). We have re-written the paragraph (ln. 100-104) to make this more clear.

[Figure]
 **Comment R3.28**

Equation 1: Shouldn't CON (used in the text) and H2O be non-italic?

**Response to R3.28:** Thanks for noticing this. Amended as suggested.

[Figure]
 **Comment R3.29**

Equation 3: Symbol "h" not introduced.

**Response to R3.29:** $|\Delta h|$ defined as the Lower Heating Value.

[Figure]
 **Comment R3.30**

L121: bar on c_p is not placed appropriately.

**Response to R3.30:** Overline fixed as suggested.

[Figure]
 **Comment R3.31**

L123: "define **it** to be"

**Response to R3.31:** This section has been rewritten.
* * *
L129: "written **as**"

**Response to R3.32:** Rewritten to "[...], or sometimes $\Theta$ [...]" (ln. 142).
* * *
✎ **Comment R3.33**

L130: "is tangent to" would be a more exact description, I think

**Response to R3.33:** Amended as suggested (ln. 143).
* * *
✎ **Comment R3.34**

Equation 5: Units here are written using italics whereas that is not the case in the main text.

**Response to R3.34:** Thank for noticing. Units amended to non-italic case.
* * *
✎ **Comment R3.35**

L151: "to limit **required** computational resources" ?

**Response to R3.35:** Amended as suggested.
* * *
✎ **Comment R3.36**

L153: "last" or "latest" ? Or "previous" ?

**Response to R3.36:** We have changed "last generation" to "previous generation" as suggested.
* * *
✎ **Comment R3.37**

L160: What are the implications of ignoring liquid droplets in the plume?

**Response to R3.37**: Thanks for catching this mistake. What we meant to say is that we ignore the emission of stored, liquid water that is released outside of ice supersaturated regions. This is the goal of water vapour extraction technologies. But it is obviously not the same as ignoring liquid droplets in the exhaust plume. We have amended this sentence accordingly (ln. 174–176).
* * *
✏ **Comment R3.38**

Table 1: Abbreviation "JA1" not introduced

**Response to R3.38**: Amended to "CON".
* * *
✏ **Comment R3.39**

Table 1: Why not just state some of the Q values as MJ/kg?

**Response to R3.39**: Amended as suggested.
* * *
✏ **Comment R3.40**

Table 1: G missing units.

**Response to R3.40**: Added units.
* * *
✏ **Comment R3.41**

L189: "We use numpy.random.choice" Without replacement I presume?

**Response to R3.41**: Indeed, we have now specified "without replacement".
* * *
✏ **Comment R3.42**

L201: What are "further ERA5 variables"? Perhaps better to say "variables other than the relative humidity" ?

**Response to R3.42**: Amended as suggested.
* * *
✏ **Comment R3.43**

L236: "in the horizontal direction" ?

**Response to R3.43**: Amended as suggested.

**Comment R3.44**

Figure 2: What is "limfac edge" in the colormap label?

**Response to R3.44:** Removed "limfac" abbreviation.

**Comment R3.45**

L259: "Saved to file" can just be "saved" right?

**Response to R3.45:** Amended as suggested.

**Comment R3.46**

L299: "acts like" these are different physical processes, right? Perhaps different wording might be more appropriate?

**Response to R3.46:** Thanks for this comment. We replaced "acts" with "behaves". We understand that this is personification of physical processes, but we believe this is the most understandable way of getting our point across.

**Comment R3.47**

L311: "in **the** horizontal direction"?

**Response to R3.47:** Amended as suggested.

**Comment R3.48**

L312: "conducive to persistent contrail formation" Shouldn't this be "for all ambient conditions satisfying persistence and droplet freezing"?

**Response to R3.48:** This sentence has been removed.

**Comment R3.49**

L321: "in **the** vertical direction"

**Response to R3.49:** Amended as suggested.

✐ **Comment R3.50**

L330: Are there any examples of this happening?

**Response to R3.50:** We are unfortunately unable to quantify this with our present study, which is why we only mention that this is what the results "suggest". We have added a following statement that further work would be required to prove this theory.

✐ **Comment R3.51**

L351: "conservatively regrid" What does this mean?

**Response to R3.51:** By "conservatively regrid", we simply mean that we preserve the total integral (sum) of the input field in the output. This is important in our case, since we would like the total flown distance to remain the same. We have added "conserving the total sum of distance flown" (ln. 371) to make this clearer.

✐ **Comment R3.52**

L363: Why the number of varying significant digits reported?

**Response to R3.52:** We amended the WET-50 significant digits to match others within the paragraph. The second significant digit for 99.96 % was left since it would be misleading to round this to 100 %.

✐ **Comment R3.53**

L368: Why the negative latitudes if you already specify that they are on the Southern hemisphere?

**Response to R3.53:** Removed the minus signs.

✐ **Comment R3.54**

Figure 6: I suggest using different colors for the lines here as one might confuse the fact that they represent different pressure levels with differing aircraft designs. Perhaps use colors from a sequential colormap? Also, no need to have ".0" behind all the pressure levels in the legend.

**Response to R3.54:** We do not believe the colours need to be changed since the legend clearly shows the values. We have removed the decimal places and added "hPa" to the pressure level values in the legend.

✐ **Comment R3.55**

Table 3: I suggest to use italics for the symbols.

**Response to R3.55:** Amended as suggested.
* * *
✏️ **Comment R3.56**

L456: This sentence is quite hard to read, consider splitting it up.

**Response to R3.56:** Thanks for this suggestion. Rephrased to "As contrails spread vertically and/or horizontally over time, they may persist in grid cells that our analysis identified as limited by droplet formation."
* * *
✏️ **Comment R3.57**

L473: "towards to poles" should be "towards the poles" ?

**Response to R3.57:** Amended as suggested.
* * *
**References**

Agarwal, A., Meijer, V. R., Eastham, S. D., Speth, R. L., and Barrett, S. R. H.: Reanalysis-driven simulations may overestimate persistent contrail formation by 100%–250%, Environ. Res. Lett., 17, 014045, https://doi.org/10.1088/1748-9326/ac38d9, 2022.

Bier, A., Unterstrasser, S., and Vancassel, X.: Box model trajectory studies of contrail formation using a particle-based cloud microphysics scheme, Atmos. Chem. Phys., 22, 823–845, https://doi.org/10.5194/acp-22-823-2022, 2022.

Bier, A., Unterstrasser, S., Zink, J., Hillenbrand, D., Jurkat-Witschas, T., and Lottermoser, A.: Contrail formation on ambient aerosol particles for aircraft with hydrogen combustion: a box model trajectory study, Atmos. Chem. Phys., 24, 2319–2344, https://doi.org/10.5194/acp-24-2319-2024, 2024.

Brock, C. A., Froyd, K. D., Dollner, M., Williamson, C. J., Schill, G., Murphy, D. M., Wagner, N. J., Kupc, A., Jimenez, J. L., Campuzano-Jost, P., Nault, B. A., Schroder, J. C., Day, D. A., Price, D. J., Weinzierl, B., Schwarz, J. P., Katich, J. M., Wang, S., Zeng, L., Weber, R., Dibb, J., Scheuer, E., Diskin, G. S., DiGangi, J. P., Bui, T., Dean-Day, J. M., Thompson, C. R., Peischl, J., Ryerson, T. B., Bourgeois, I., Daube, B. C., Commane, R., and Wofsy, S. C.: Ambient aerosol properties in the remote atmosphere from global-scale in situ measurements, Atmos. Chem. Phys., 21, 15023–15063, https://doi.org/10.5194/acp-21-15023-2021, 2021.

Geraedts, S., Brand, E., Dean, T. R., Eastham, S., Elkin, C., Engberg, Z., Hager, U., Langmore, I., McCloskey, K., Yue-Hei Ng, J., Platt, J. C., Sankar, T., Sarna, A., Shapiro, M., and Goyal, N.: A scalable system to measure contrail formation on a per-flight basis, Environ. Res. Commun., 6, 015008, https://doi.org/10.1088/2515-7620/ad11ab, 2024.

Gierens, K.: Theory of Contrail Formation for Fuel Cells, Aerospace, 8, 164, https://doi.org/10.3390/aerospace8060164, 2021.

Haywood, J. M., Allan, R. P., Bornemann, J., Forster, P. M., Francis, P. N., Milton, S., Rädel, G., Rap, A., Shine, K. P., and Thorpe, R.: A case study of the radiative forcing of persistent contrails evolving into contrail-induced cirrus, J. Geophys. Res., 114, 2009JD012650, https://doi.org/10.1029/2009JD012650, 2009.

Hildebrandt, K. G.: Ice Supersaturated Regions: Validation of ERA5 Reanalysis with IAGOS In-Situ Measurements and Effect on Contrail Formation Potential of Flights, Delft University of Technology, Delft, NL, 2024.

Hofer, S., Gierens, K., and Rohs, S.: How well can persistent contrails be predicted? An update, Atmos. Chem. Phys., 24, 7911–7925, https://doi.org/10.5194/acp-24-7911-2024, 2024.

Immler, F., Treffeisen, R., Engelbart, D., Krüger, K., and Schrems, O.: Cirrus, contrails, and ice supersaturated regions in high pressure systems at northern mid latitudes, Atmos. Chem. Phys., 8, 1689–1699, https://doi.org/10.5194/acp-8-1689-2008, 2008.

Kärcher, B., Burkhardt, U., Bier, A., Bock, L., and Ford, I. J.: The microphysical pathway to contrail formation, J. Geophys. Res. Atmos., 120, 7893–7927, https://doi.org/10.1002/2015JD023491, 2015.

Kärcher, B.: Formation and radiative forcing of contrail cirrus, Nature Communications, 9, 1824–1824, https://doi.org/10.1038/s41467-018-04068-0, 2018.

Reutter, P., Neis, P., Rohs, S., and Sauvage, B.: Ice supersaturated regions: properties and validation of ERA-Interim reanalysis with IAGOS in situ water vapour measurements, Atmos. Chem. Phys., 20, 787–804, https://doi.org/10.5194/acp-20-787-2020, 2020.

Schumann, U.: On conditions for contrail formation from aircraft exhausts, Meteorologische Zeitschrift, 5, 4–23, https://doi.org/10.1127/metz/5/1996/4, 1996.

Schumann, U. and Heymsfield, A. J.: On the Life Cycle of Individual Contrails and Contrail Cirrus, Meteorological Monographs, 58, 3.1–3.24, https://doi.org/10.1175/AMSMONOGRAPHS-D-16-0005.1, 2017.

Wolf, K., Bellouin, N., Boucher, O., Rohs, S., and Li, Y.: Correction of temperature and relative humidity biases in ERA5 by bivariate quantile mapping: Implications for contrail classification, https://doi.org/10.5194/egusphere-2023-2356, 9 November 2023.

Wolf, K., Bellouin, N., and Boucher, O.: Distribution and morphology of non-persistent and persistent contrail formation areas in ERA5, https://doi.org/10.5194/egusphere-2023-3086, 11 January 2024.

Wolf, K., Bellouin, N., Boucher, O., Rohs, S., and Li, Y.: Correction of ERA5 temperature and relative humidity biases by bivariate quantile mapping for contrail formation analysis, Atmos. Chem. Phys., 25, 157–181, https://doi.org/10.5194/acp-25-157-2025, 2025.

Voigt, C., Lelieveld, J., Schlager, H., Schneider, J., Curtius, J., Meerkötter, R., Sauer, D., Bugliaro, L., Bohn, B., Crowley, J. N., Erbertseder, T., Groß, S., Hahn, V., Li, Q., Mertens, M., Pöhlker, M. L., Pozzer, A., Schumann, U., Tomsche, L., Williams, J., Zahn, A., Andreae, M., Borrmann, S., Bräuer, T., Dörich, R., Dörnbrack, A., Edtbauer, A., Ernle, L., Fischer, H., Giez, A., Granzin, M., Grewe, V., Harder, H., Heinritzi, M., Holanda, B. A., Jöckel, P., Kaiser, K., Krüger, O. O., Lucke, J., Marsing, A., Martin, A., Matthes, S., Pöhlker, C., Pöschl, U., Reifenberg, S., Ringsdorf, A., Scheibe, M., Tadic, I., Zauner-Wieczorek, M., Henke, R., and Rapp, M.: Cleaner Skies during the COVID-19 Lockdown, Bulletin of the American Meteorological Society, 103, E1796–E1827, https://doi.org/10.1175/BAMS-D-21-0012.1, 2022.

Yin, F., Grewe, V., and Gierens, K.: Impact of Hybrid-Electric Aircraft on Contrail Coverage, Aerospace, 7, 147, https://doi.org/10.3390/aerospace7100147, 2020.